# Time-PEFT: Temporal and Multichannel Complexity-Based Fine-Tuning for Time-Series Foundation Models

Jihye Na [1]  Patara Trirat [2]  Chanyoung Park [3]  Jae-Gil Lee [1]

## Abstract

Recent studies have attempted to fine-tune time-series foundation models to enhance a target dataset's forecasting performance. However, these approaches proceed without a clear criterion for identifying complex datasets that require fine-tuning due to performance degradation in zero-shot forecasting. To distinguish datasets that are more challenging than standard benchmarks, we introduce data-driven temporal complexity and multichannel complexity. *Temporal complexity* captures the difficulty of identifying distinct patterns by quantifying spectral entropy in the frequency domain, while *multichannel complexity* captures cross-channel information flow that can impact predictive uncertainty. These metrics serve as *effective proxies for performance gains* achievable through fine-tuning. Based on the two metrics, we develop *Time-PEFT*, a parameter-efficient fine-tuning framework that incorporates a frequency adapter for top-$k$ filtering and a channel adapter for multichannel modeling. With the base variant of MOMENT as a backbone, *Time-PEFT* improves performance by up to 38% over LoRA on complex datasets.

## 1. Introduction

Recent studies have investigated fine-tuning strategies for time-series foundation models (TSFMs) to improve task-specific performance beyond the universal representations acquired during pretraining. Representative methods, including LoRA (Hu et al., 2022), AdaPTS (Benechehab et al., 2025), and MSFT (Qiao et al., 2025), report consistent gains

on standard forecasting benchmarks (e.g., electricity transformer temperature and weather datasets). However, we find that the improvements on more complex datasets (e.g., medical and chaotic datasets) are often limited. Notably, since these complex datasets exhibit less overlap with the pretraining data than the standard datasets, one would expect fine-tuning to yield larger performance gains. Contrary to this expectation, the observed improvements remain similar because of the inability of generic fine-tuning methods to adequately model the intrinsic complexity of such datasets, including sophisticated temporal dynamics and cross-channel information patterns. These findings underscore the need for fine-tuning approaches for TSFMs that can effectively accommodate a *wide range of data complexities*.

To this end, we propose a complexity-aware parameter-efficient fine-tuning (PEFT) framework, *Time-PEFT*, specifically designed to enhance forecasting performance of TSFMs on complex datasets. Our approach comprises two key components: (i) the formulation of data complexity metrics that characterize two sources of difficulty, and (ii) the development of parameter-efficient adaptation mechanisms tailored to address each identified complexity.

**Data Complexity Formulation**: Viewing a multivariate time series as a two-dimensional structure, we characterize dataset complexity for forecasting along two axes: the time axis and the channel (variable) axis, as shown in Figure 1 (a). Accordingly, we define temporal complexity and multichannel complexity to capture the intrinsic difficulty of modeling temporal dynamics and cross-channel information, respectively. *Temporal complexity* is derived from spectral entropy, which represents how uniformly the power spectral density is distributed across the frequency domain to indicate inherent difficulty in identifying distinct patterns within the time series. *Multichannel complexity* is computed via transfer entropy, which quantifies the reduction in uncertainty of the target channel's future by incorporating the history of another channel given its own history, thereby capturing complex information flows among multiple channels.

A key advantage is that these two complexity metrics serve as dataset-level indicators of when complexity-aware fine-tuning is likely to yield larger gains. For temporal and multichannel complexity, a generic PEFT method such as

[1]School of Computing, Korea Advanced Institute of Science & Technology, Daejeon, South Korea [2]DeepAuto.ai, Seoul, South Korea [3]Industrial and Systems Engineering, Korea Advanced Institute of Science & Technology, Daejeon, South Korea. Correspondence to: Jae-Gil Lee <jaegil@kaist.ac.kr>.

*Proceedings of the 43rd International Conference on Machine Learning*, Seoul, South Korea. PMLR 306, 2026. Copyright 2026 by the author(s).

**Figure 1. Illustration of the two axes of multivariate time series.** (a) Conceptual visualization of standard and complex datasets in terms of temporal and multichannel complexity. For temporal complexity, complex datasets exhibit uniform frequency distributions. For multichannel complexity, complex datasets depict dataset-level channel information flow. (b) A comparison between generic PEFT methods, which are not designed for temporal and multichannel complexity, and *Time-PEFT*. For high temporal complexity, fine-tuning should focus on identifying distinct patterns. For high multichannel complexity, fine-tuning should address multichannel modeling.

LoRA (Hu et al., 2022) exhibits limited performance improvement as complexity increases. In contrast, the proposed PEFT method tends to achieve progressively larger gains with increasing complexity. Consequently, the performance gap between generic approaches and our method widens as dataset complexity grows.

**Complexity Handling**: Since we observe that data complexity is the primary challenge faced by generic PEFT methods, a PEFT approach tailored for complex datasets should effectively address each type of complexity. *Temporal complexity* stems from the absence of clear periodic patterns. In such datasets, a dominant frequency often does not exist, and instead the frequency spectrum is dispersed, causing the frequency components to behave like noise and undermining predictive accuracy. *Multichannel complexity*, in contrast, arises from dataset-level cross-channel information flow, indicating that information from one channel can reduce uncertainty about another channel. Despite the importance of handling complexity, generic PEFT methods are not designed to address both types of complexity simultaneously, as illustrated in Figure 1(b-1). Specifically, on the time axis, generic PEFT methods struggle to generalize on complex datasets due to the sparse temporal patterns. Meanwhile, on the channel axis, generic PEFT methods perform channel-specific processing without a dedicated channel-common component for multichannel modeling.

To effectively address multichannel complexity and temporal complexity, we introduce a *channel adapter* and a *frequency adapter*, respectively. The frequency adapter identifies and emphasizes the top-$k$ dominant frequency components to alleviate temporal complexity, while the channel adapter learns channel-common representations to mitigate the effects of multichannel complexity. More specifically, as shown in Figure 1(b-2), the frequency adapter emphasizes clear temporal patterns by focusing on dominant frequencies; the channel adapter encourages channel-common knowledge via shared weights while extracting individual channel knowledge through respective channel weights in a parameter-efficient manner.

**Performance Gains**: Comprehensive experiments with *five* TSFMs on *seven* complex datasets demonstrate that our proposed *Time-PEFT* achieves competitive or superior performance compared to *five* fine-tuning baselines. In particular, with the base variant of MOMENT (Goswami et al., 2024) as the backbone TSFM, *Time-PEFT* achieves up to 38% lower error than LoRA. These findings underscore that substantial performance enhancements can be realized by adding a small number of trainable parameters.

## 2. Related Work

### 2.1. Time-Series Foundation Models (TSFMs)

The number of studies about TSFMs has increased due to zero-shot forecasting capabilities after the advent of OFA (Zhou et al., 2023). Pre-training TSFMs presents several challenges. The first challenge arises from the lack of time-series datasets compared to other modalities. To address this data scarcity, TimesFM (Das et al., 2024) utilized large-scale synthetic time-series datasets, while other models, such as Timer (Liu et al., 2024b) and Chronos (Ansari et al., 2024), leveraged large-scale collections of diverse time-series datasets. Another challenge is effectively incorporating the intrinsic properties of time-series data during the pre-training phase. Regarding the channel aspect, unlike univariate forecasting paradigms adopted by MOMENT (Goswami et al., 2024) and Sundial (Liu et al., 2025c), UniTS (Gao et al., 2024a) applies self-attention across both temporal and channel dimensions for multivariate forecasting. In the temporal aspect, recent advancements focus on extended contexts, as seen in Timer-XL (Liu et al., 2025b), or capturing varying densities through multi-patch size strategies, as proposed in Moirai (Woo et al., 2024). Meanwhile, some studies focus on model scalability. Guided by scaling laws that performance correlates with the number of model parameters, Time-MoE (Shi et al., 2025) aims to increase model capacity. Conversely, there is a shift towards lightweight TSFMs, such as Tiny Time Mixers (Ekambaram et al., 2024).

*Table 1.* Comparisons between related works in terms of multi-channel and temporal aspects.

| Fine-Tuning Method | Channel | Temporal |
|---|:---:|:---:|
| LoRA (Hu et al., 2022) | ✗ | ✗ |
| PCD (Lee et al., 2024) | ✓ | ✗ |
| Channel Mixing (Ekambaram et al., 2024) | ✓ | ✗ |
| AdaPTS (Benechehab et al., 2025) | ✓ | ✗ |
| MSFT (Qiao et al., 2025) | ✗ | ✓ |
| *Time-PEFT* (Ours) | ✓ | ✓ |

### 2.2. Fine-Tuning for TSFMs

Recently, distinct from general fine-tuning methods (e.g., Low-Rank Adaptation (LoRA) (Hu et al., 2022)), specialized fine-tuning strategies tailored for TSFMs have emerged. These methods account for intrinsic time-series characteristics, such as channel or temporal aspects, which are not present in other modalities. However, as summarized in Table 1, previous works have *not* addressed both aspects simultaneously. Furthermore, the majority of prior studies evaluate their methods only on standard forecasting benchmarks, such as ETT datasets, and often neglect to identify specific datasets where existing zero-shot forecasting performs poorly on TSFMs. In contrast, our study not only considers both channel and temporal aspects but also identifies datasets exhibiting a significant performance gap between zero-shot inference and fine-tuned forecasting.

Representative studies that fine-tune TSFMs from a channel perspective include PCD (Lee et al., 2024), ChannelMixing (Ekambaram et al., 2024), and AdaPTS (Benechehab et al., 2025). PCD fine-tunes partial channel dependence but applies only to TSFMs that have variable-wise input tokens, thereby lacking generalizability to diverse architectures. ChannelMixing implicitly captures channel dependence by transposing the time and channel dimensions with fully connected layers. AdaPTS adopts various adapter-based architectures to adapt univariate TSFMs for multivariate forecasting. On the other hand, in terms of the temporal aspect, MSFT (Qiao et al., 2025) captures multi-scale patterns. Additionally, Gen-P-Tuning (Liu et al., 2025a), a prompt-tuning-inspired method, is evaluated in the healthcare domain rather than on ETT datasets.

## 3. Data Complexity Analysis

### 3.1. Definitions

To quantify the complexity of datasets, which are often challenging for zero-shot TSFMs, we introduce data-driven definitions for two types of complexity. Temporal complexity captures the forecasting difficulty arising from the absence of dominant periodic patterns (Goerg, 2013; Ponce-Flores et al., 2020); a high value implies that signal energy is

distributed uniformly across frequencies. Orthogonally, multichannel complexity measures the strength of information flow between channels, identifying cases where multichannel modeling is necessary.

**Definition 3.1** (TEMPORAL COMPLEXITY). *Temporal complexity* ($S_{\text{temp}}$) quantifies the average normalized spectral entropy (SE) (Shannon, 1948; Inouye et al., 1991). The metric is defined by normalizing spectral entropy by $\log_2 N_f$ and averaging across all $C$ channels, where $N_f$ denotes the number of frequency bins. Formally,

$$S_{\text{temp}} = \frac{1}{C} \sum_{i=1}^{C} \frac{\text{SE}_i}{\log_2 N_f}, \tag{1}$$

where the spectral entropy of the $i$-th channel is defined as

$$\text{SE}_i = -\sum_{k=1}^{N_f} p_i(f_k) \log_2 p_i(f_k). \tag{2}$$

Here, the power spectral density, $P_i(f_k)$, for the $i$-th channel is normalized by its sum to form a probability distribution $p_i(f_k)$, where $f_k$ represents the $k$-th frequency component.

**Definition 3.2** (MULTICHANNEL COMPLEXITY). *Multichannel complexity* ($S_{\text{ch}}$) quantifies the cross-channel information contribution using time-delayed transfer entropy (TE) (Schreiber, 2000; Wibral et al., 2013; Sipahi & Porfiri, 2020). To capture the maximum coupling strength regardless of the interaction, we define the final score as the average of the maximum TE values across a look-back window $\mathcal{T} = \{1, \ldots, L\}$. Here,

$$S_{\text{ch}} = \frac{1}{C(C-1)} \sum_{i=1}^{C} \sum_{\substack{j=1 \\ j \neq i}}^{C} \max_{\tau \in \mathcal{T}} \left( \text{TE}_{j \to i}(\tau) \right). \tag{3}$$

Specifically, the TE from the $j$-th channel to the $i$-th channel with a time lag $\tau$ is defined as

$$\begin{aligned}\text{TE}_{j \to i}(\tau) = \ & H(X_t^{(i)}|X_{t-1}^{(i)}) \\ & - H(X_t^{(i)}|X_{t-1}^{(i)}, X_{t-\tau}^{(j)}),\end{aligned} \tag{4}$$

where $X_t^{(i)}$ denotes the $i$-th channel time series at time step $t$, and $H(\cdot|\cdot)$ denotes the conditional entropy (Cover & Thomas, 1991).

Unlike linear correlation, TE measures the directed information flow by calculating the reduction in uncertainty of the target $i$-th channel time series given the history of the source $j$-th channel time series.

### 3.2. Identifying Complex Datasets for Fine-Tuning

We empirically define *complex datasets* as those exhibiting high temporal complexity ($> 0.6$) or high multichannel complexity ($> 0.1$); these complexities serve as primary factors

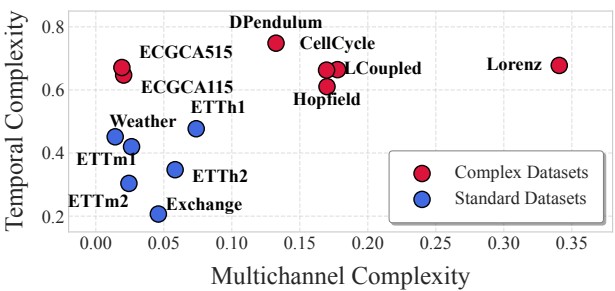

*Figure 2.* Positioning of complex and standard datasets with respect to temporal and multichannel complexity.

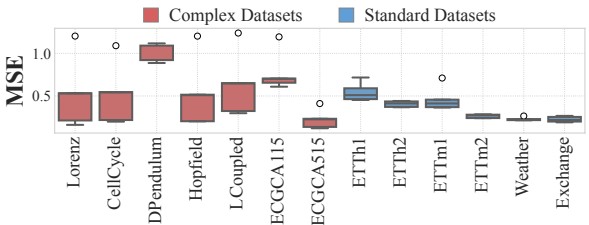

*Figure 3.* Average mean squared error (MSE) comparison between complex and standard datasets among zero-shot results and existing four fine-tuning methods on UniTS. Full comparisons with other TSFMs are provided in Appendix A.

limiting zero-shot performance. In Figure 2, standard benchmark datasets have low temporal complexity below 0.6 and low multichannel complexity below 0.1. In contrast, complex datasets have high temporal complexity and varying multichannel complexity. To show the need for fine-tuning TSFMs on complex datasets, Figure 3 presents a boxplot for complex and standard datasets across zero-shot inference and fine-tuning strategies (forecast head fine-tuning, full fine-tuning, LoRA (Hu et al., 2022), and FourierFT (Gao et al., 2024b)). For complex datasets, the zero-shot MSE is substantially higher in most cases, appearing as upper outliers relative to fine-tuning results. Furthermore, these results exhibit considerably larger variance. Conversely, standard datasets demonstrate a narrower variance in the boxplots, indicating that the absolute performance disparity between zero-shot inference and fine-tuning methods is marginal. Therefore, the high zero-shot MSE and the large variance of performance gains highlight the need for fine-tuning strategies tailored to complex datasets.

As illustrated in Figure 4(a) for temporal complexity and Figure 4(b) for multichannel complexity, *Time-PEFT*, which has additional time-series adapters, exhibits a performance advantage over LoRA, particularly on datasets characterized by high complexity. These results suggest that generic PEFT approaches are insufficient for complex datasets in TSFMs. Our findings indicate that such time-series-specific adaptation becomes increasingly indispensable as the temporal and multichannel complexities of the dataset rise.

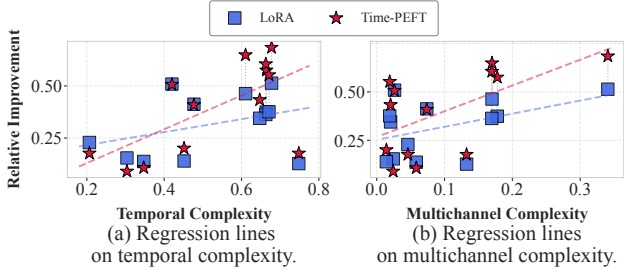

*Figure 4.* Relative performance improvements of LoRA (blue) and *Time-PEFT* (red) over the zero-shot MOMENT$_{\text{base}}$.

## 4. Method: *Time-PEFT*

### 4.1. Problem Formulation

Given a model $M_{\text{pre}}$ pre-trained on a pre-training dataset $S_{\text{pre}}$, our goal is to adapt this model to an unseen target dataset. Therefore, the performance of the fine-tuned model $M_{\text{fine}}$ from a fine-tuning dataset $S_{\text{fine}} \in \{X_t\}_{t=1}^{N_t}$ can be better than the pre-trained model. Let $\mathbf{X} \in \mathbb{R}^{[B \times C \times T]}$ denote multivariate time-series inputs, where $B$ is the batch size, $C$ is the number of channels, and $T$ is the look-back window length. Given the input time series, the purpose of multivariate time-series forecasting is to predict the future time series $\mathbf{Y} \in \mathbb{R}^{[B \times C \times L]}$ with forecast horizon $L$.

### 4.2. Overview of *Time-PEFT*

Our proposed method, *Time-PEFT*, comprises a frequency adapter and a channel adapter integrated with PEFT mechanisms as illustrated in Figure 5. First, we apply PEFT while keeping the original pre-trained backbone frozen to preserve pre-trained knowledge. Next, the frequency adapter extracts filtered representations from the backbone embedding (§4.3). These filtered representations are then combined with the original backbone embedding and fed into the channel adapter (§4.4). Finally, channel embeddings are forwarded to the forecast head for the final prediction. Refer to Appendix B for our algorithm.

### 4.3. Temporal Complexity Handling

#### 4.3.1. THEORETICAL MOTIVATION

Prior literature establishes that increased entropy increases the forecasting difficulty (Ponce-Flores et al., 2020; Garland et al., 2014). To mitigate temporal complexity, we propose top-$k$ filtering that clarifies time-series patterns by retaining only the dominant top-$k$ frequency components. From an information-theoretic perspective, predictability is intrinsically linked to the spectral properties of time series. The Kolmogorov-Szegő formula suggests that, specifically under stationary Gaussian assumptions, a process with a peaky spectral density is associated with a lower entropy rate and

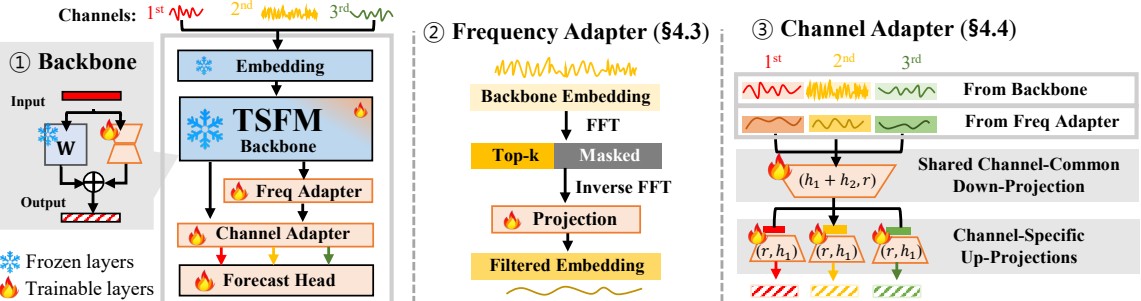

*Figure 5.* Architecture of *Time-PEFT* for multivariate time-series forecasting. The architecture has three key components for effectively adapting TSFMs: (1) The backbone network is fine-tuned using PEFT methods such as LoRA (Hu et al., 2022). (2) The frequency adapter extracts a filtered embedding enriched with filtered patterns from the backbone embedding. (3) Based on the embeddings from both the backbone and the frequency adapter, channel embeddings are generated and subsequently fed into the forecast head.

thus smaller irreducible prediction error $\sigma^2$ compared to a uniform spectrum (Cover & Thomas, 1991). Since top-$k$ filtering can reduce the entropy rate, under the assumptions, $\sigma^2_{\text{filt}} < \sigma^2_{\text{raw}}$, with the prediction error of the filtered input $\sigma^2_{\text{filt}}$ and that of raw input $\sigma^2_{\text{raw}}$. Although satisfying these assumptions requires strict conditions, this formulation theoretically supports that the prediction error bound can be reduced through filtering. See Appendix C.1 for more details. However, aggressive filtering poses a risk of discarding salient intrinsic patterns by treating them as noise, potentially rendering the performance sensitive to the hyperparameter $k$. To mitigate this potential loss from filtering, *Time-PEFT* uses both the filtered features via the frequency adapter and the original backbone embedding.

### 4.3.2. FREQUENCY ADAPTER (FIGURE 5(2))

Since the frequency domain provides an effective representation for capturing temporal patterns (Na et al., 2025), many studies have explored time-series dynamics through dominant top-$k$ frequency components (Wu et al., 2023; Kang et al., 2025). Motivated by these studies, we employ top-$k$ filtering so that the frequency adapter can recover distinct patterns and obtain the filtered embedding by masking less significant frequency components while retaining only the top-$k$ frequencies through the temporal filtering mechanism. Formally, given the output embedding $\mathbf{E}_{\text{back}} \in \mathbb{R}^{[B \times C \times K \times h_1]}$ from the last layer of the backbone, where $h_1$ denotes the backbone's hidden dimension and $K$ is the number of temporal patches in each TSFM, the filtered embedding $\mathbf{E}_{\text{filt}} \in \mathbb{R}^{[B \times C \times K \times h_2]}$ is computed as

$$\mathbf{E}_{\text{filt}} = \text{Proj}(\text{IFFT}(\text{TopK}(\text{FFT}(\mathbf{E}_{\text{back}})))). \quad (5)$$

Here, $\text{FFT}(\cdot)$ denotes the fast Fourier transform along the patch dimension, converting the patch-level representation into the frequency domain. The $\text{TopK}(\cdot)$ selects the $k$ frequency components with the largest amplitudes and masks the remaining ones. $\text{IFFT}(\cdot)$ represents the inverse fast Fourier transform, which reconstructs the filtered signals

back into the patch domain. Given the filtered temporal representation, $\text{Proj}(\cdot)$ is a learnable linear layer that maps the hidden dimension from $h_1$ to $h_2$, which is the output embedding dimension of the frequency adapter. To retain access to the original information, the filtered and original embeddings are fed together into the channel adapter.

### 4.4. Multichannel Complexity Handling

#### 4.4.1. THEORETICAL MOTIVATION

While Fano's inequality formally bounds the error probability in discrete identification tasks, we use it as an information-theoretic motivation under a quantized view of forecasting. In this view, reducing the conditional entropy of the target can lower the Fano-derived error-bound term. Motivated by the transfer entropy in Equation 4, we introduce two terms for the following multichannel discussion. We refer to information derived from the target channel history $X^{(i)}_{t-1}$ as *channel-specific information*, and information from another channel's history $X^{(j)}_{t-\tau}$ that reduces target uncertainty as *channel-common information*. We compare two idealized prediction regimes. The first is *specific-only* (SO) prediction, which uses only channel-specific information. The second is *specific-and-common* (SC) prediction, which uses channel-specific information together with channel-common information. Let $P^{\text{SO}}_e$ and $P^{\text{SC}}_e$ denote the corresponding error probabilities. Let $\mathcal{B}(P_e)$ represent the theoretical lower bound of the error probability $P_e$ derived from Fano's inequality (Cover & Thomas, 1991). When another channel history provides positive information gain, the lower-bound term for SC prediction is no larger than that for SO prediction: $\mathcal{B}(P^{\text{SC}}_e) \leq \mathcal{B}(P^{\text{SO}}_e)$. A larger information gain derived from $X^{(j)}_{t-\tau}$ suggests a more substantial reduction in the Fano-derived bound. Consequently, this provides motivation for multichannel modeling on datasets with high multichannel complexity, where channel-common information is likely to reduce the target channel's uncertainty. Appendix C.2 provides detailed theoretical motivation.

#### 4.4.2. CHANNEL ADAPTER (FIGURE 5(3))

Motivated by the *specific-and-common* prediction view, we design the channel adapter for multichannel modeling. The adapter adds a channel-common component on top of channel-specific processing through shared and channel-specific parameterization. A shared *channel-common* down-projection matrix $W^{(S)} \in \mathbb{R}^{[(h_1+h_2) \times r]}$ is applied to each channel with the same parameters, encouraging reusable latent representations via parameter sharing. The *channel-specific* up-projection matrices specialize these representations for each channel. Given the $c$-th channel backbone embedding $\mathbf{E}_{\text{back}}^{(c)}$ and filtered embedding $\mathbf{E}_{\text{filt}}^{(c)}$, the $c$-th channel embedding is computed as

$$\mathbf{E}_{\text{ch}}^{(c)} = \text{Drop}(\text{ReLU}([\mathbf{E}_{\text{back}}^{(c)} \| \mathbf{E}_{\text{filt}}^{(c)}]W^{(S)}))W^{(c)}, \quad (6)$$

where the $c$-th channel-specific up-projection is $W^{(c)} \in \mathbb{R}^{[r \times h_1]}$, $r$ is the adapter rank, and $\text{ReLU}(\cdot)$ and $\text{Drop}(\cdot)$ denote the rectified linear unit and dropout, respectively. Finally, $\mathbf{E}_{\text{ch}}^{(c)}$ is normalized and passed to the forecast head.

### 4.5. Parameter Analysis of *Time-PEFT*

Our framework augments existing PEFT methods by integrating frequency and channel adapters. Both adapters use projections via fully connected layers. To simplify the analysis, we examine the number of parameters excluding bias and normalization terms. First, the frequency adapter consists of a single projection layer; thus, its parameter count is given by $h_1 \times h_2$. The channel adapter comprises one shared down-projection and $C$ channel-specific up-projections. Consequently, the total number of parameters for the channel adapter is calculated as $r \times (h_1 + h_2) + C \times r \times h_1$. In summary, the total model size in *Time-PEFT* is primarily determined by the backbone dimension $h_1$ and the number of channels $C$. Regarding scaling behavior, the parameter count grows linearly with $h_1$, with a per-dimension growth rate of $O(h_2 + (C + 1) \times r)$. Meanwhile, each additional channel increases the parameter count by $O(r \times h_1)$.

## 5. Experiments

### 5.1. Experiment Settings

#### 5.1.1. COMPLEX DATASETS

To evaluate performance on complex datasets, we use the simulated time-series datasets from chaotic systems (Gilpin, 2021). We select the five chaotic systems that have the largest Lyapunov exponents (Abdelmalak et al., 2025): **Lorenz**, **CellCycle**, DoublePendulum (**DPendulum**), **Hopfield**, and LorenzCoupled (**LCoupled**). These datasets are generated from mathematical models spanning diverse fields, including meteorology, biology, classical mechanics, and neuroscience. For real-world evaluation,

we use medical datasets, which are reported as challenging for TSFMs (Gupta et al., 2024a;b). We employ two representative non-invasive fetal ECG datasets from PhysioNet (Goldberger et al., 2000): **ECGCA115** and **ECGCA515**. Appendix D.1 provides detailed dataset descriptions and experimental setup.

#### 5.1.2. BACKBONE TSFMs AND IMPLEMENTATION

We follow the widely adopted codebase and experimental setup for time-series forecasting as implemented in TSLib (Wang et al., 2026) with a look-back window of 96 and long-term forecast horizons of 96, 192, and 336. We use five pre-trained TSFMs as backbones: **MOMENT$_{\text{small}}$**, **MOMENT$_{\text{base}}$** (Goswami et al., 2024), **UniTS** (Gao et al., 2024a), **Chronos** (Ansari et al., 2024), and **TTM** (Ekambaram et al., 2024). For the encoder-decoder models Chronos and TTM, only their encoders are utilized as backbones. Except for TTM, which requires a look-back window of 512 due to its pre-training constraint restricting supported inputs to 512 or 1024, we maintain a look-back window of 96. The forecasting results are evaluated using Mean Absolute Error (MAE) and Mean Squared Error (MSE). All experiments are conducted under a single NVIDIA RTX 4090 budget. The training process includes early stopping with a maximum of 100 epochs.

#### 5.1.3. COMPARISON BASELINES

We compare *Time-PEFT* against several fine-tuning approaches: zero-shot forecasting without any tuning (**Zeroshot**); fine-tuning only the forecast head of TSFMs (**HeadOnly**); full fine-tuning of all parameters (**FullFT**); other parameter-efficient fine-tuning methods such as **LoRA** (Hu et al., 2022) and **FourierFT** (Gao et al., 2024b); and other time-series fine-tuning methods such as **ChannelMixing** (Ekambaram et al., 2024). To adapt ChannelMixing for diverse TSFMs, we incorporate it into TSFMs as a final layer of the encoder. Except for the zero-shot setting, all other baselines fine-tune the forecast head of TSFMs. Furthermore, we include end-to-end models, PatchTST (Nie et al., 2023) and iTransformer (Liu et al., 2024a), as additional baselines.

### 5.2. Main Results

As presented in Table 2, zero-shot forecasting substantially exhibits performance degradation on complex datasets, thereby highlighting the necessity of fine-tuning. On average, *Time-PEFT* achieves superior performance across diverse TSFMs. For MOMENT$_{\text{base}}$, our method outperforms all TSFM fine-tuning baselines across complex datasets. For MOMENT$_{\text{small}}$, UniTS, Chronos$_{\text{small}}$, and TTM$_{\text{r2}}$, *Time-PEFT* achieves the best performance on most datasets. Meanwhile, full fine-tuning often ranks among the top-

*Table 2.* Average MAE and MSE for long-term forecasting across forecast horizons {96, 192, 336} on complex datasets. Within each TSFM block, the best and second-best results are shown in **bold** and underlined, respectively; gray rows denote zero-shot results.

| Method | Lorenz MAE | Lorenz MSE | CellCycle MAE | CellCycle MSE | DPendulum MAE | DPendulum MSE | Hopfield MAE | Hopfield MSE | LCoupled MAE | LCoupled MSE | ECGCA115 MAE | ECGCA115 MSE | ECGCA515 MAE | ECGCA515 MSE |
|---|---|---|---|---|---|---|---|---|---|---|---|---|---|---|
| PatchTST | 0.522 | 0.559 | 0.489 | 0.612 | 0.844 | 0.970 | 0.586 | 0.597 | 0.643 | 0.720 | 0.484 | 0.699 | 0.278 | 0.241 |
| iTransformer | 0.580 | 0.641 | 0.445 | 0.532 | 0.839 | 0.956 | 0.492 | 0.474 | 0.695 | 0.789 | 0.459 | 0.633 | 0.266 | 0.236 |
| **MOMENT**small | 0.910 | 1.238 | 0.759 | 1.119 | 0.915 | 1.134 | 0.950 | 1.234 | 0.910 | 1.263 | 0.777 | 1.207 | 0.429 | 0.418 |
| + HeadOnly | 0.616 | 0.697 | 0.581 | 0.784 | 0.854 | 0.990 | 0.686 | 0.733 | 0.734 | 0.897 | 0.561 | 0.809 | 0.303 | 0.270 |
| + Full FT | 0.434 | **0.415** | 0.441 | 0.530 | **0.806** | **0.906** | 0.621 | 0.615 | 0.533 | 0.534 | 0.541 | 0.768 | 0.313 | 0.277 |
| + LoRA | 0.574 | 0.635 | 0.558 | 0.732 | 0.846 | 0.977 | 0.650 | 0.680 | 0.702 | 0.838 | 0.559 | 0.806 | 0.300 | 0.266 |
| + FourierFT | 0.580 | 0.643 | 0.562 | 0.741 | 0.847 | 0.979 | 0.656 | 0.689 | 0.706 | 0.846 | 0.559 | 0.806 | 0.300 | 0.266 |
| + ChannelMixing | 0.611 | 0.687 | 0.538 | 0.685 | 0.850 | 0.988 | 0.611 | 0.639 | 0.668 | 0.762 | 0.546 | 0.749 | 0.280 | 0.231 |
| + *Time-PEFT* | **0.420** | 0.428 | **0.430** | **0.514** | 0.813 | 0.918 | **0.481** | **0.461** | **0.524** | **0.534** | 0.459 | **0.670** | **0.251** | **0.190** |
| **MOMENT**base | 0.903 | 1.220 | 0.755 | 1.104 | 0.910 | 1.113 | 0.946 | 1.218 | 0.906 | 1.251 | 0.776 | 1.203 | 0.428 | 0.415 |
| + HeadOnly | 0.588 | 0.659 | 0.569 | 0.751 | 0.851 | 0.985 | 0.666 | 0.714 | 0.716 | 0.867 | 0.550 | 0.792 | 0.297 | 0.264 |
| + Full FT | 0.470 | 0.465 | 0.481 | 0.599 | 0.817 | 0.923 | 0.629 | 0.631 | 0.553 | 0.566 | 0.559 | 0.809 | 0.327 | 0.294 |
| + LoRA | 0.541 | 0.594 | 0.538 | 0.691 | 0.843 | 0.971 | 0.625 | 0.654 | 0.677 | 0.797 | 0.547 | 0.788 | 0.294 | 0.259 |
| + FourierFT | 0.550 | 0.606 | 0.544 | 0.704 | 0.844 | 0.974 | 0.634 | 0.667 | 0.685 | 0.811 | 0.548 | 0.788 | 0.294 | 0.259 |
| + ChannelMixing | 0.631 | 0.734 | 0.530 | 0.673 | 0.856 | 1.001 | 0.613 | 0.649 | 0.667 | 0.761 | 0.527 | 0.721 | 0.266 | 0.215 |
| + *Time-PEFT* | **0.385** | **0.385** | **0.405** | **0.469** | **0.807** | **0.916** | **0.452** | **0.428** | **0.492** | **0.494** | 0.459 | **0.682** | **0.248** | **0.186** |
| **UniTS** | 0.897 | 1.203 | 0.752 | 1.090 | 0.904 | 1.091 | 0.942 | 1.203 | 0.902 | 1.239 | 0.773 | 1.194 | 0.427 | 0.411 |
| + HeadOnly | 0.500 | 0.532 | 0.456 | 0.546 | 0.814 | 0.922 | 0.531 | 0.509 | 0.609 | 0.651 | 0.505 | 0.704 | 0.267 | 0.225 |
| + Full FT | **0.200** | **0.161** | 0.219 | 0.199 | 0.797 | 1.118 | 0.262 | 0.202 | **0.280** | 0.323 | 0.432 | 0.702 | 0.203 | 0.137 |
| + LoRA | 0.239 | 0.212 | 0.241 | 0.217 | 0.766 | 0.887 | 0.255 | 0.203 | 0.301 | 0.298 | 0.414 | 0.609 | 0.197 | 0.121 |
| + FourierFT | 0.499 | 0.530 | 0.452 | 0.542 | 0.814 | 0.922 | 0.534 | 0.515 | 0.607 | 0.648 | 0.475 | 0.655 | 0.272 | 0.232 |
| + ChannelMixing | 0.533 | 0.574 | 0.530 | 0.703 | 0.832 | 0.956 | 0.657 | 0.719 | 0.660 | 0.755 | 0.581 | 0.947 | 0.270 | 0.225 |
| + *Time-PEFT* | 0.207 | 0.177 | **0.173** | **0.137** | **0.761** | **0.869** | **0.223** | **0.171** | 0.290 | **0.275** | 0.399 | 0.548 | **0.195** | **0.119** |
| **Chronos**small | 0.900 | 1.397 | 0.776 | 1.359 | 1.074 | 1.798 | 0.953 | 1.409 | 0.895 | 1.306 | 0.794 | 1.791 | 0.439 | 0.643 |
| + HeadOnly | 0.645 | 0.738 | 0.655 | 0.800 | 0.849 | 0.969 | 0.649 | 0.712 | 0.759 | 0.858 | 0.649 | 0.884 | 0.390 | 0.320 |
| + Full FT | 0.292 | 0.370 | 0.401 | 0.522 | **0.764** | 0.874 | 0.442 | 0.467 | 0.439 | 0.532 | 0.710 | 1.240 | 0.483 | 0.541 |
| + LoRA | 0.489 | 0.544 | 0.597 | 0.760 | 0.831 | 0.960 | 0.578 | 0.653 | 0.648 | 0.736 | 0.753 | 1.210 | 0.506 | 0.604 |
| + FourierFT | 0.645 | 0.737 | 0.656 | 0.802 | 0.849 | 0.969 | 0.649 | 0.712 | 0.759 | 0.859 | 0.649 | 0.885 | 0.390 | 0.319 |
| + ChannelMixing | 0.675 | 0.819 | 0.606 | 0.824 | 0.878 | 1.069 | 0.661 | 0.727 | 0.750 | 0.936 | 0.629 | 0.904 | 0.357 | 0.321 |
| + *Time-PEFT* | **0.284** | **0.286** | **0.394** | **0.440** | 0.775 | **0.864** | **0.407** | **0.377** | **0.412** | **0.416** | 0.533 | 0.715 | 0.316 | 0.232 |
| **TTM**r2 | 0.910 | 1.265 | 0.789 | 1.212 | 0.948 | 1.274 | 0.862 | 1.108 | 0.841 | 1.092 | 0.694 | 1.125 | 0.382 | 0.394 |
| + HeadOnly | 0.671 | 0.798 | 0.586 | 0.737 | 0.852 | 0.978 | 0.638 | 0.650 | 0.743 | 0.844 | 0.512 | 0.685 | 0.277 | 0.236 |
| + Full FT | 0.509 | 0.538 | 0.443 | 0.501 | 0.830 | **0.940** | 0.490 | 0.449 | 0.610 | 0.634 | 0.422 | **0.513** | 0.226 | 0.160 |
| + LoRA | 0.566 | 0.625 | 0.489 | 0.570 | 0.841 | 0.954 | 0.525 | 0.493 | 0.663 | 0.713 | 0.440 | 0.544 | 0.237 | 0.179 |
| + FourierFT | 0.671 | 0.798 | 0.586 | 0.737 | 0.852 | 0.978 | 0.638 | 0.650 | 0.743 | 0.844 | 0.512 | 0.685 | 0.277 | 0.236 |
| + ChannelMixing | 0.682 | 0.804 | 0.614 | 0.729 | 0.853 | 0.969 | 0.619 | 0.621 | 0.739 | 0.817 | 0.535 | 0.660 | 0.300 | 0.218 |
| + *Time-PEFT* | **0.438** | **0.472** | **0.357** | **0.372** | **0.827** | 0.958 | **0.383** | **0.334** | **0.521** | **0.550** | 0.403 | 0.533 | **0.211** | **0.140** |

performing baselines on complex datasets for the four TSFMs. Considering that full fine-tuning demands significant trainable parameters, *Time-PEFT* remains a competitive and efficient alternative. Overall, these results demonstrate that *Time-PEFT* is an effective and parameter-efficient solution tailored for complex datasets. Refer to Table 6 in Appendix D.2.1 for detailed results on complex datasets.

In addition to Figure 2, Figure 6 visualizes the relative improvement of *Time-PEFT* over LoRA on diverse datasets. Here, the marker sizes reflect the magnitude of improvement for MOMENTbase. It is noticeable that the gains from *Time-PEFT* tend to scale with the temporal and multichannel complexity of a dataset. While high-complexity datasets yield significant improvements (big red circles), these gains

diminish in lower-complexity datasets (small blue circles), showing that **our data complexity metrics** (Equation 1 and Equation 3) **provide useful indicators of the potential improvements** when fine-tuned with *Time-PEFT*.

Although *Time-PEFT* is primarily designed for complex datasets, we also report its performance on standard datasets. Appendix D.2.2 presents the overall results on standard datasets. Table 3 summarizes the range of relative average MSE improvements achieved by fine-tuning methods over LoRA, categorized by dataset type. These results show that the gains of *Time-PEFT* are most pronounced on complex datasets, while its performance on standard datasets remains comparable to existing baselines, suggesting *Time-PEFT* can be applied across different levels of data complexity.

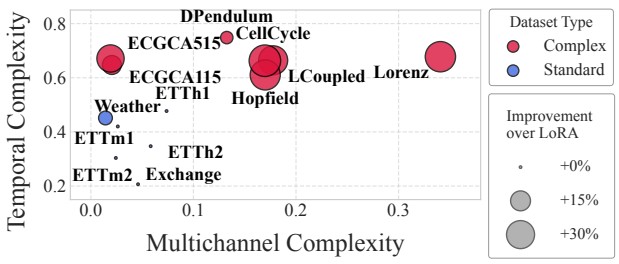

*Figure 6.* Performance improvement with respect to complexity. The size of markers represents the relative MSE improvement (%) of *Time-PEFT* compared to LoRA on MOMENT$_{base}$.

*Table 3.* Relative improvement (%) comparison between *Time-PEFT* and baselines for the two types of datasets on MOMENT$_{base}$.

| Datasets | FourierFT | ChannelMixing | *Time-PEFT* |
|---|---|---|---|
| Complex | $\{-2.1 \sim -0.1\}$ | $\{-23.6 \sim 17.0\}$ $\ll$ | $\{\mathbf{5.6 \sim 38.0}\}$ |
| Standard | $\{-0.8 \sim 0.2\}$ | $\{-27.7 \sim -3.8\}$ $\approx$ | $\{-7.6 \sim 7.1\}$ |

*Table 4.* Average MSE in terms of combinations of modules using MOMENT$_{base}$: **L** (LoRA), **C** (Channel adapter), and **F** (Frequency adapter). We distinguish two variants of **F**: $\mathbf{F_{only}}$ transmits only the filtered embedding to the subsequent module, whereas $\mathbf{F_{both}}$ propagates both the filtered and backbone embeddings.

| | **Fine-Tuning Modules** | | | | |
|---|---|---|---|---|---|
| **Datasets** | L | L+C | L+F$_{only}$ | L+F$_{both}$ | L+C+F$_{both}$ |
| Lorenz | 0.594 | 0.417 | 0.544 | 0.520 | **0.385** |
| CellCycle | 0.691 | 0.491 | 0.665 | 0.627 | **0.469** |
| DPendulum | 0.971 | 0.917 | 0.971 | 0.961 | **0.916** |
| Hopfield | 0.654 | 0.451 | 0.619 | 0.601 | **0.428** |
| LCoupled | 0.797 | 0.526 | 0.733 | 0.705 | **0.494** |
| ECGCA115 | 0.788 | 0.688 | 0.750 | 0.745 | **0.682** |
| ECGCA515 | 0.259 | 0.204 | 0.245 | 0.238 | **0.186** |

## 5.3. Ablation Study and Sensitivity Analysis

### 5.3.1. EFFECTS OF PROPOSED ADAPTERS

Table 4 presents the performance variations when integrating our proposed modules into LoRA applied to MOMENT$_{base}$. The channel adapter and frequency adapter yield a performance improvement compared to LoRA. $F_{only}$, which uses only the filtered embeddings, yields a slight performance improvement. Meanwhile, $F_{both}$, which retains the original backbone embeddings together with the filtered embeddings, demonstrates better performance with the existing embeddings. As a result, fine-tuning both adapters achieves the best performance. We attribute this success to the channel-specific modules within the channel adapter, which enable the distinct modeling of dominant frequency patterns for individual channels. Consequently, the synergistic effect of both adapters enables *Time-PEFT* (L+C+F$_{both}$) to operate effectively.

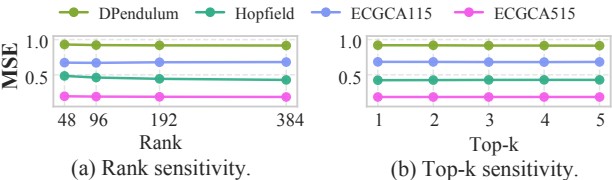

*Figure 7.* Hyperparameter sensitivity analysis in terms of rank $r$ and top-$k$ frequency components on MOMENT$_{base}$.

*Table 5.* Average number of trainable parameters and their percentages on the Lorenz dataset using MOMENT$_{base}$.

| Methods | # Params (Millions) | Percentage (%) |
|---|---|---|
| Full FT | 111.553 | 100.000 |
| HeadOnly | 1.917 | 1.719 |
| LoRA | 2.360 | 2.115 |
| FourierFT | 1.917 | 1.719 |
| ChannelMixing | 1.917 | 1.719 |
| *Time-PEFT* | 4.429 | 3.970 |

### 5.3.2. HYPERPARAMETER SENSITIVITY

We investigate the sensitivity to the hyperparameters, the adapter rank $r$ and the number of top-$k$ frequency components on MOMENT$_{base}$, as shown in Figure 7. All main experiments for our method are conducted with default configurations of $r = h_1/2$ and $k = 3$; for MOMENT$_{base}$, $r = 384$ because $h_1 = 768$. For the adapter rank, we evaluate ranks from 48 to 384. The results show that the performance is not highly sensitive to the choice of $r$. We also vary $k$ from 1 to 5 to control the number of dominant frequency components along the temporal patches. The results are similarly robust to the choice of $k$.

### 5.3.3. PARAMETER EFFICIENCY

Table 5 presents a comparison of the number of trainable parameters across selected fine-tuning baselines and *Time-PEFT* under the default adapter rank setting $r = h_1/2$ on the Lorenz dataset using MOMENT$_{base}$. Despite the inclusion of both channel and frequency adapters, our method exhibits a small increase in trainable parameters compared to the standard LoRA. Substantial performance enhancements on complex datasets can be achieved with small parameter overhead by incorporating specialized time-series fine-tuning. Moreover, the smaller-rank variant with $r = h_1/16$, corresponding to $r = 48$, requires only 3.138M trainable parameters (2.813%), indicating that *Time-PEFT* can be scaled down when parameter efficiency is prioritized.

## 6. Conclusion

This paper introduces two entropy-based metrics, *temporal* and *multichannel* complexity, to identify challenging

datasets where TSFM adaptation is likely to be beneficial. Guided by this complexity analysis, we propose *Time-PEFT*, a complexity-aware parameter-efficient fine-tuning method for TSFMs that integrates a channel adapter for multichannel modeling and a frequency adapter equipped with top-$k$ filtering to extract complementary temporal representations in multivariate time series. Experiments demonstrate that *Time-PEFT* achieves competitive or superior adaptation performance compared with existing fine-tuning baselines across complex datasets and backbone models.

## Acknowledgements

This work was supported by Institute of Information & Communications Technology Planning & Evaluation (IITP) grant funded by the Korea government (MSIT) (No. RS-2020-II200862, DB4DL: High-Usability and Performance In-Memory Distributed DBMS for Deep Learning, 50% and No. RS-2022-II220157, Robust, Fair, Extensible Data-Centric Continual Learning, 50%).

## Impact Statement

This paper aims to advance TSFMs by introducing efficient adaptation strategies. Analogous to how fine-tuning strategies in natural language processing have enabled researchers and practitioners to leverage general knowledge for customized downstream applications, we envision that our approach will facilitate a similar evolution in time series analysis. By enabling lightweight and time-series-specific adaptation, this work may enhance forecasting capabilities across diverse time-series scenarios.

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

## A. The Necessity for Fine-Tuning TSFMs

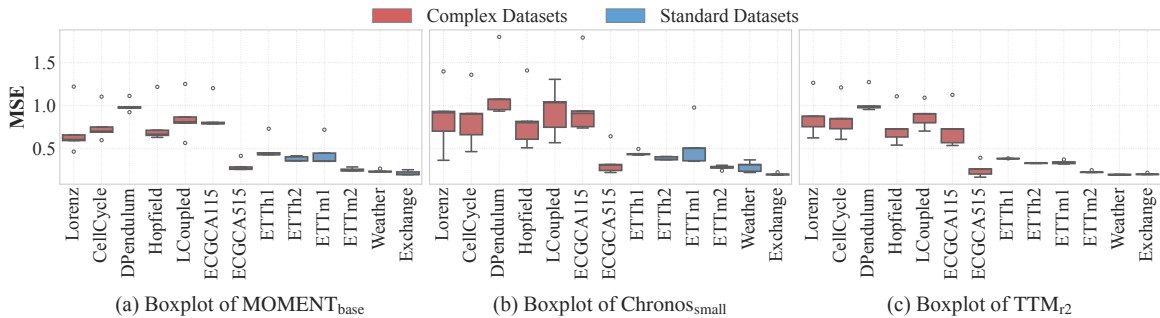

*Figure 8.* Average MSE comparison between complex and standard datasets among zero-shot inference and existing four fine-tuning methods (forecast head fine-tuning, full fine-tuning, LoRA, and FourierFT) on MOMENT$_{\text{base}}$, Chronos$_{\text{small}}$, and TTM$_{\text{r2}}$. Note that most upper outliers correspond to zero-shot results except for TTM's full fine-tuning on the ETTh1 and exchange datasets.

Figure 8 extends the analysis presented in Figure 3 for additional TSFMs. The results are consistent with our previous findings; complex datasets on each TSFM exhibit larger variability in MSE, and the zero-shot results on complex datasets are usually upper outliers relative to the fine-tuned results. Moreover, TSFMs generally demonstrate poorer intrinsic zero-shot performance on complex datasets compared to that on standard datasets. Overall, the performance gap between zero-shot inference and fine-tuned models shows the need for fine-tuning on complex datasets.

## B. Algorithm

---

**Algorithm 1** Overall Fine-Tuning Procedure of *Time-PEFT*

---

INPUT: Target Multivariate Time-series Dataset $S_{\text{fine}} = \{(\mathbf{X}, \mathbf{Y})\}$, Pre-trained Backbone $M_{\text{pre}}$
INPUT: Hyperparameters: Adapter rank $r$, Top-$k$ frequency components $k$, Number of Epochs $E$, Learning Rate $\eta$.
OUTPUT: Optimized Parameters $\Theta^*$.
 1: **Freezing:** Apply LoRA to pre-defined layers of $M_{\text{pre}}$ while keeping other layers of $M_{\text{pre}}$ frozen except forecast head.
 2: **Initialize:** Trainable parameters $\Theta \leftarrow \{\theta_{\text{LoRA}}, \theta_{\text{Freq}}, \theta_{\text{Chan}}, \theta_{\text{Head}}\}$.
 3: **for** $epoch = 1$ **to** $E$ **do**
 4:    **for** each batch $(\mathbf{X}, \mathbf{Y})$ from $S_{\text{fine}}$ **do**
 5:       /* Step 1: Backbone with LoRA */
 6:       $\mathbf{E}_{\text{back}} \leftarrow M_{\text{pre}}(\mathbf{X}; \theta_{\text{LoRA}})$                               ▷ *Obtain Backbone Embeddings*
 7:       /* Step 2: Frequency Adapter */
 8:       $\mathbf{F} \leftarrow \text{FFT}(\mathbf{E}_{\text{back}})$                                 ▷ *Transform to Frequency Domain*
 9:       $\mathbf{F}_{\text{topk}} \leftarrow \text{TopK}(\mathbf{F}, k)$                             ▷ *Select Dominant Top-k Frequencies*
10:       $\mathbf{E}_{\text{filt}} \leftarrow \text{Proj}(\text{IFFT}(\mathbf{F}_{\text{topk}}))$                       ▷ *Obtain Filtered Embeddings*
11:       /* Step 3: Channel Adapter */
12:       **for** each channel $c \in \{1, \ldots, C\}$ **do**
13:          $\mathbf{Z}^{(c)} \leftarrow [\mathbf{E}_{\text{back}}^{(c)} \| \mathbf{E}_{\text{filt}}^{(c)}]$             ▷ *Concatenate Backbone and Filtered Embeddings*
14:          $\mathbf{h}^{(c)} \leftarrow \text{Drop}(\text{ReLU}(\mathbf{Z}^{(c)} \mathbf{W}^{(S)}))$         ▷ *Use Shared Down-Projection*
15:          $\mathbf{E}_{\text{ch}}^{(c)} \leftarrow \mathbf{h}^{(c)} \mathbf{W}^{(c)}$               ▷ *Use Channel-Specific Up-Projection*
16:       **end for**
17:       /* Step 4: Forecast Head */
18:       $\hat{\mathbf{Y}} \leftarrow \text{ForecastHead}(\text{LayerNorm}(\mathbf{E}_{\text{ch}}))$
19:       $\mathcal{L} \leftarrow \text{MSE}(\hat{\mathbf{Y}}, \mathbf{Y})$                                 ▷ *Calculate Loss*
20:       $\Theta \leftarrow \Theta - \eta \nabla_\Theta \mathcal{L}$                            ▷ *Update Parameters*
21:    **end for**
22: **end for**
23: **Return** $\Theta^*$

---

Algorithm 1 details the procedural execution of our fine-tuning framework, corresponding to the visual architecture illustrated in Figure 5. *Time-PEFT* is controlled by two primary hyperparameters: the rank $r$ and the frequency threshold $k$, which denotes the number of retained top-$k$ frequency components. The objective of this framework is to fine-tune the frequency and channel adapters integrated into existing PEFT mechanisms. Assuming LoRA as the default underlying PEFT method, our approach optimizes the following set of learnable parameters: $\theta_{\text{LoRA}}$ (low-rank adaptation for the pre-defined layers of the backbone), $\theta_{\text{Freq}}$ (frequency adapter), $\theta_{\text{Chan}}$ (channel adapter), and $\theta_{\text{Head}}$ (forecast head).

# C. Theoretical Background and Motivation

In this study, we establish the entropy-based theoretical motivation underlying the design of the two modules in *Time-PEFT*. We present theorems in a condensed form to highlight core insights. For details, we refer to Equations 11.39 and 11.40 for the Kolmogorov-Szegő formula and Equation 2.146 for Fano's inequality in Cover & Thomas (1991).

### C.1. Theoretical Motivation for Top-$k$ Filtering

We provide the information-theoretic motivation for top-$k$ filtering. Leveraging the Kolmogorov-Szegő (Cover & Thomas, 1991), we suggest that modifying spectral entropy via top-$k$ filtering can intuitively reduce errors under idealized assumptions.

**Theorem C.1** (Kolmogorov-Szegő formula with entropy rate). *For a stationary Gaussian process $\mathcal{X}$ with power spectral density $P(f)$, the entropy rate $h(\mathcal{X})$ is related to the Kolmogorov-Szegő formula and the irreducible innovation variance (one-step-ahead prediction error) $\sigma_\infty^2$ as follows (Cover & Thomas, 1991):*

$$\sigma_\infty^2 = \frac{1}{2\pi e} e^{2h(\mathcal{X})}, \tag{7}$$

*where the entropy rate is given by*

$$h(\mathcal{X}) = \frac{1}{2} \ln 2\pi e + \frac{1}{4\pi} \int_{-\pi}^{\pi} \ln P(f) \, df. \tag{8}$$

C.1.1. THEORETICAL MOTIVATION VIA KOLMOGOROV-SZEGŐ FORMULA

Let $\mathcal{X} = \{X_t\}_{t=-\infty}^{\infty}$ denote the stochastic Gaussian process for an arbitrary time-series channel. In this section, as we address an arbitrary channel, we omit channel-specific notations (e.g., superscripts in $X_t^{(i)}$). Under the constraint of fixed total power, the Kolmogorov-Szegő formula implies that uniform power spectral density (flat $P(f)$) maximizes entropy rate $h(\mathcal{X})$, analogous to white noise. Consequently, a flatter spectrum maximizes the internal term in Eq. 8, thereby exponentially increasing $\sigma_\infty^2$. In this idealized setting, $\sigma_\infty^2$ corresponds to the irreducible one-step-ahead error variance.

Based on this, we provide the intuition for error reduction via top-$k$ filtering. Let $\mathcal{X}_{\text{filt}}$ represent the process after top-$k$ filtering, where $P(f)$ is modified to be sparse, containing only $k$ dominant peaks in the frequency domain. Instead of the maximum-entropy uniform distribution, the filtered spectrum becomes peaky. This redistribution is expected to reduce the effective spectral entropy of the representation by concentrating energy on dominant frequencies. Although strict hard-thresholding renders the integral undefined, under the assumption of effective noise suppression, we use the following relation as an idealized motivation for a potentially reduced error bound:

$$h(\mathcal{X}_{\text{filt}}) < h(\mathcal{X}) \implies \sigma_{\text{filt}}^2 < \sigma_{\text{raw}}^2, \tag{9}$$

where $\sigma_{\text{filt}}^2$ and $\sigma_{\text{raw}}^2$ denote the irreducible error variances of the filtered and raw processes, respectively. Consequently, by applying top-$k$ filtering, we enable datasets characterized by high $S_{\text{temp}}$ to yield filtered embeddings under the idealized assumptions in C.1.2. Thus, this may reduce the error bound. In contrast, for datasets inherently possessing low $S_{\text{temp}}$, the error reduction attributable to filtering may be negligible due to their pre-existing low spectral entropy. Furthermore, caution is warranted, as aggressive filtering poses a risk of discarding latent patterns essential for forecasting performance.

C.1.2. ASSUMPTIONS: STATIONARY GAUSSIAN PROCESS

Although real-world datasets generally exhibit non-linear and non-stationary dynamics, we approximate the data within a finite window as a locally stationary Gaussian process. This assumption is justified on two grounds:

(i) *Stationarity:* The local dynamics within a short look-back window can be effectively approximated as stationary (Dahlhaus, 1997). Although assuming strict stationarity over the entire horizon is challenging due to distribution shifts

in long-term time-series forecasting, we assume quasi-stationarity of the spectral properties. Specifically, the dominant frequencies identified in the look-back window are assumed to persist or vary slowly over the forecasting horizon.

(ii) *Gaussianity:* For a fixed covariance, the Gaussian distribution yields the maximum entropy rate (Cover & Thomas, 1991). Therefore, deriving the error bound under the Gaussian assumption provides a conservative reference point for analyzing inherent uncertainty, justifying our objective to minimize this bound.

## C.2. Theoretical Motivation for Multichannel Modeling

We provide a theoretical motivation for multichannel modeling, leveraging Fano's inequality (Cover & Thomas, 1991). We show that, under a quantized forecasting view, incorporating uncertainty-reducing information from other channels can lower the uncertainty term appearing in the Fano-derived error bound compared with specific-only prediction.

**Theorem C.2** (Fano's Inequality)**.** *Let $U$ be a discrete target random variable defined on a finite set $\mathcal{U}$, and let $V$ be an observed random variable correlated with $U$. Consider an estimator $g$ that attempts to predict $U$ based on the observation $V$ (i.e., $\hat{U} = g(V)$). Let $P_e = P(\hat{U} \neq U)$ denote the probability of error. Fano's inequality upper bounds the conditional entropy $H(U|V)$ in terms of the error probability, and equivalently provides a lower bound on the error probability as follows:*

$$H(P_e) + P_e \log(|\mathcal{U}| - 1) \geq H(U|V), \tag{10}$$

*where $H(P_e) = -P_e \log P_e - (1 - P_e) \log(1 - P_e)$ is the binary entropy function. For simplicity, a weak form of Fano's inequality is often utilized to bound the error probability:*

$$P_e \geq \frac{H(U|V) - 1}{\log |\mathcal{U}|}. \tag{11}$$

*This inequality implies that the minimum error rate is bounded by the remaining uncertainty of the target $U$ given $V$.*

### C.2.1. THEORETICAL MOTIVATION VIA FANO'S INEQUALITY

To derive the Fano-derived error-bound terms for *specific-only* and *specific-and-common* prediction, we adapt the simplified Fano's inequality (Eq. 11). Since Fano's inequality applies to discrete variables, we consider a quantized forecasting task. We define the theoretical lower bound of the decision error probability, denoted as $\mathcal{B}(P_e)$. First, we formalize the variables for the forecasting context. Let $X_t^{(i)}$ denote the target variable of the $i$-th channel at time $t$, and let $X_{<t}$ represent the available history that is observed up to time $t - 1$. To accommodate the discrete nature of Fano's inequality, we assume continuous variables $X$ and $X_{<t}$ are quantized into discrete $|\mathcal{Q}|$ bins for entropy estimation. Consequently, the lower bound for the error probability $P_e$ for the target $X_t^{(i)}$ given input $X_{<t}$ is formulated as:

$$P_e \geq \frac{H(X_t^{(i)}|\mathbf{X_{<t}}) - 1}{\log |\mathcal{Q}|}. \tag{12}$$

In *specific-only* (SO) prediction, the prediction relies solely on the historical data of the target channel. Consistent with our definition of transfer entropy on multichannel complexity in Equation 3, we adopt the first-order Markov assumption, considering the immediate observation $X_{t-1}^{(i)}$ as the dominant history. Under the assumption, the error lower bound for SO prediction is given by:

$$\mathcal{B}(P_e^{\text{SO}}) = \frac{H(X_t^{(i)}|X_{t-1}^{(i)}) - 1}{\log |\mathcal{Q}|}. \tag{13}$$

Conversely, in *specific-and-common* (SC) prediction, the prediction exploits historical data from both the target channel and another channel. Thus, incorporating the observation from the $j$-th channel at a lag of $\tau$ steps $X_{t-\tau}^{(j)}$ alongside $X_{t-1}^{(i)}$, the error lower bound for SC prediction is:

$$\mathcal{B}(P_e^{\text{SC}}) = \frac{H(X_t^{(i)}|X_{t-1}^{(i)}, X_{t-\tau}^{(j)}) - 1}{\log |\mathcal{Q}|}. \tag{14}$$

Recall that channel complexity is derived from transfer entropy (TE) as defined in Equation 4. By definition, TE represents the reduction in uncertainty and corresponds to the difference between the conditional entropy of the SO prediction and that

of the SC prediction. Therefore, the reduction in the Fano-derived lower bound on the decision-error probability gained by incorporating information from another channel is quantified as:

$$\mathcal{B}(P_e^{\text{SO}}) - \mathcal{B}(P_e^{\text{SC}}) = \frac{H(X_t^{(i)}|X_{t-1}^{(i)}) - H(X_t^{(i)}|X_{t-1}^{(i)}, X_{t-\tau}^{(j)})}{\log |\mathcal{Q}|}$$
$$= \frac{\text{TE}_{j \to i}(\tau)}{\log |\mathcal{Q}|}. \tag{15}$$

Since $\text{TE}_{j \to i}(\tau) \geq 0$ holds and the quantization bin size $|\mathcal{Q}|$ is a constant, this yields the following relation between the Fano-derived error-bound terms:

$$\mathcal{B}(P_e^{\text{SO}}) \geq \mathcal{B}(P_e^{\text{SC}}). \tag{16}$$

This result provides an information-theoretic motivation for multichannel modeling under the idealized assumptions, such as a quantized view and a one-step prediction. A larger transfer entropy indicates that channel-common information can reduce more uncertainty about the target channel, suggesting a stronger motivation for using multichannel modeling on datasets with high $S_{\text{ch}}$. Conversely, when $S_{\text{ch}}$ is low, the Fano-derived bound gap is small, suggesting that additional multichannel modeling may provide limited benefit.

# D. Detailed Experiments

## D.1. Details of Experimental Setup

### D.1.1. DETAILED DESCRIPTION OF COMPLEX DATASETS

In this study, our evaluation focuses on complex datasets, comprising five chaotic systems and two medical datasets. Regarding the chaotic systems, Abdelmalak et al. (2025) identified six systems characterized by the largest Lyapunov exponents, highlighting their channel dependencies. Building upon this study, we selected five chaotic systems to measure the complexity of datasets exhibiting diverse inter-channel relationships. As these chaotic systems are synthetic, we generated the datasets using the benchmark codebase (Gilpin, 2021). Additionally, the medical datasets are non-invasive fetal ECG recordings from PhysioNet (Goldberger et al., 2000). A brief description of each dataset is provided below.

- **Lorenz** is a mathematical model for atmospheric convection, consisting of 3 channels.
- **CellCycle** represents the growth and division processes in biological systems, consisting of 6 channels.
- **DPendulum** (DoublePendulum) involves pendulum dynamics, comprising 4 channels.
- **Hopfield** is a dataset modeling state transitions within neural networks, consisting of 6 channels.
- **LCoupled** (LorenzCoupled) is formed by coupling multiple Lorenz systems, comprising a total of 6 channels.
- **ECGCA115** and **ECGCA515** consist of 6 channels with 2 thoracic signals and 4 abdominal signals.

### D.1.2. IMPLEMENTATION DETAILS

In this study, we employ a diverse set of five TSFMs to evaluate fine-tuning performance on complex datasets. Our evaluation encompasses encoder-based TSFMs, such as MOMENT$_{\text{small}}$, MOMENT$_{\text{base}}$, and UniTS. Meanwhile, for encoder-decoder architectures, we employ Chronos$_{\text{small}}$ and TTM$_{\text{r2}}$. Implementation details for each model are provided below:

- **MOMENT$_{\text{small}}$ and MOMENT$_{\text{base}}$**: We use the pre-trained weights from `AutonLab/MOMENT-1-small, base`.
- **UniTS**: Experiments are conducted using the pre-trained weights of the backbone in the checkpoint of GitHub (`units_x32_pretrain_checkpoint.pth`).
- **Chronos$_{\text{small}}$**: We adopt the pre-trained weights from `amazon/chronos-bolt-small`. Since Chronos follows an encoder-decoder architecture, our main experiments use the encoder as the backbone for fine-tuning baselines. Meanwhile, zero-shot evaluation uses the original encoder-decoder model. For the boxplot in Figure 8(b), we additionally report the original encoder-decoder baselines (Zero-shot, HeadOnly, FullFT, LoRA, and FourierFT) to reflect their standard usage.
- **TTM$_{\text{r2}}$**: We use the pre-trained `ibm-granite/granite-timeseries-ttm-r2` model. Since TTM is an encoder-decoder architecture, our main experiments use the encoder for architectural consistency. Meanwhile, zero-shot evaluation uses the original encoder-decoder model. For the boxplot in Figure 8(c), we additionally report the original encoder-decoder baselines following the same policy as Chronos. In addition, we set the look-back window to 512 exclusively for the TTM experiments. This is because TTM's pre-training weights support only lengths of 512 or 1024.

D.1.3. HYPERPARAMETER SETTINGS FOR FINE-TUNING METHODS

We benchmark the performance of *Time-PEFT* against a zero-shot baseline and five fine-tuning strategies. All experiments are implemented based on TSLib (Wang et al., 2026), using a maximum of 100 epochs with early stopping. The learning rate is selected within the range between $10^{-3}$ and $10^{-5}$ with the AdamW optimizer. We use the default batch size of 128 and adjust it for certain combinations of datasets, backbone models, and training strategies under the computational budget of a single NVIDIA RTX 4090 GPU. Except for the zero-shot baseline, all methods update the forecast head. Thus, their modules are optimized jointly with the forecast head, while the remaining backbone parameters are kept frozen unless otherwise specified. Detailed configurations for each method are provided below:

- **Zeroshot** evaluates the pre-trained TSFM directly on the target dataset without any parameter updates or fine-tuning.
- **Full FT** updates all model parameters without freezing any components of the pre-trained TSFMs.
- **HeadOnly** updates only the parameters of the forecast head while keeping the entire backbone frozen.
- **LoRA** applies low-rank adaptation with a rank $r' = 8$ and a scale $\alpha = 32$. The strategy is tailored to the backbones. For example, for Transformer-based TSFMs, LoRA modules are injected into the query, key, and value projections of the attention mechanism. For the MLP-Mixer-based TSFMs (TTM), they are applied to the linear layers (fc1 and fc2).
- **FourierFT** operates in the frequency domain as a representative PEFT method with a small number of trainable parameters under its official default hyperparameters.
- **ChannelMixing** is originally designed for TTM to capture channel dependencies; we implement a simplified version to ensure generalization across different backbones. In our setup, a ChannelMixing module is inserted at the position designated for our adapters, referencing the TTM codebase. Note that we evaluate TTM using this modified ChannelMixing module rather than its original implementation for consistency.
- *Time-PEFT* optimizes frequency and channel adapters with LoRA modules and the forecast head. We set hyperparameters: top-$k$ is 3, adapter rank is $h_1/2$, LoRA rank is 8, and LoRA scale is 32. The backbone dimension $h_1$ follows the size of each TSFM, and we set output dimension of the frequency adapter to $h_2 = h_1$.

## D.2. Detailed Experimental Results

D.2.1. DETAILED RESULTS ON COMPLEX DATASETS

Table 6 shows the standard deviation for each TSFM and each method on complex datasets, computed across three seeds and three forecast horizons. Table 7 and Table 8 report the full results stratified by individual forecast horizons on complex datasets. As expected, error rates generally rise as the forecast horizon extends in most cases. This pattern holds across fine-tuning methods and TSFMs. For MOMENT_base, *Time-PEFT* achieves the best average performance across datasets. It remains highly competitive for MOMENT_small, UniTS, Chronos_small, and TTM_r2 as well, ranking first or second in most dataset-metric pairs.

D.2.2. DETAILED RESULTS ON STANDARD DATASETS

Table 9 shows the overall results of average MAE and MSE on standard datasets (ETTh1, ETTh2, ETTm1, ETTm2, Weather, and Exchange) across various TSFMs. While *Time-PEFT* demonstrates high performance on complex datasets, our proposed method shows more modest results on standard datasets. This result can be attributed to the low complexity of these datasets, where additional architectural components such as the frequency adapter and channel adapter may be less necessary. Consequently, conventional fine-tuning methods, including LoRA, full fine-tuning, and FourierFT, often achieve competitive or superior performance. Meanwhile, applying existing fine-tuning methods to encoder-only variants of the originally encoder-decoder-based TSFMs, such as Chronos_small and TTM_r2, can lead to performance degradation relative to zero-shot forecasting in some datasets. This vulnerability of the modified architecture is especially evident for Chronos_small on standard datasets. As illustrated in Figure 8(b), when the original Chronos_small is preserved, such degradation relative to zero-shot forecasting is not observed; instead, the baselines generally outperform the zero-shot setting. These results suggest that relying solely on the encoder may be less suitable for some standard datasets than for complex datasets.

*Table 6.* Average MAE and MSE with standard deviation for long-term forecasting across forecast horizons {96, 192, 336} and three different seeds on complex datasets. Within each TSFM block, the best and second-best fine-tuning results are shown in **bold** and underlined, respectively. Note that zero-shot results in each TSFM are colored in gray.

| Method | Lorenz MAE | Lorenz MSE | CellCycle MAE | CellCycle MSE | DPendulum MAE | DPendulum MSE | Hopfield MAE | Hopfield MSE | LCoupled MAE | LCoupled MSE | ECGCA115 MAE | ECGCA115 MSE | ECGCA515 MAE | ECGCA515 MSE |
|---|---|---|---|---|---|---|---|---|---|---|---|---|---|---|
| **MOMENTsmall** | 0.910 | 1.238 | 0.759 | 1.119 | 0.915 | 1.134 | 0.950 | 1.234 | 0.910 | 1.263 | 0.777 | 1.207 | 0.429 | 0.418 |
| + HeadOnly | 0.616 | 0.697 | 0.581 | 0.784 | 0.854 | 0.990 | 0.686 | 0.733 | 0.734 | 0.897 | 0.561 | 0.809 | 0.303 | 0.270 |
|  | ±0.07 | ±0.12 | ±0.05 | ±0.12 | ±0.02 | ±0.04 | ±0.07 | ±0.11 | ±0.07 | ±0.12 | ±0.09 | ±0.18 | ±0.05 | ±0.05 |
| + Full FT | 0.434 | **0.415** | 0.441 | 0.530 | **0.806** | **0.906** | 0.621 | 0.615 | 0.533 | 0.534 | 0.541 | 0.768 | 0.313 | 0.277 |
|  | ±0.15 | ±0.22 | ±0.14 | ±0.25 | ±0.05 | ±0.08 | ±0.10 | ±0.15 | ±0.16 | ±0.23 | ±0.10 | ±0.19 | ±0.07 | ±0.08 |
| + LoRA | 0.574 | 0.635 | 0.558 | 0.732 | 0.846 | 0.977 | 0.650 | 0.680 | 0.702 | 0.838 | 0.559 | 0.806 | 0.300 | 0.266 |
|  | ±0.07 | ±0.12 | ±0.06 | ±0.13 | ±0.03 | ±0.05 | ±0.08 | ±0.11 | ±0.07 | ±0.12 | ±0.09 | ±0.18 | ±0.05 | ±0.05 |
| + FourierFT | 0.580 | 0.643 | 0.562 | 0.741 | 0.847 | 0.979 | 0.656 | 0.689 | 0.706 | 0.846 | 0.559 | 0.806 | 0.300 | 0.266 |
|  | ±0.07 | ±0.12 | ±0.06 | ±0.13 | ±0.03 | ±0.04 | ±0.08 | ±0.11 | ±0.07 | ±0.12 | ±0.09 | ±0.18 | ±0.05 | ±0.05 |
| + ChannelMixing | 0.611 | 0.687 | 0.538 | 0.685 | 0.850 | 0.988 | 0.611 | 0.639 | 0.668 | 0.762 | 0.546 | 0.749 | 0.280 | 0.231 |
|  | ±0.06 | ±0.10 | ±0.07 | ±0.14 | ±0.03 | ±0.05 | ±0.10 | ±0.15 | ±0.09 | ±0.16 | ±0.08 | ±0.17 | ±0.05 | ±0.05 |
| + *Time-PEFT* | **0.420** | 0.428 | **0.430** | **0.514** | 0.813 | 0.918 | **0.481** | **0.461** | **0.524** | **0.534** | **0.459** | **0.670** | **0.251** | **0.190** |
|  | ±0.13 | ±0.19 | ±0.09 | ±0.18 | ±0.04 | ±0.07 | ±0.11 | ±0.14 | ±0.14 | ±0.19 | ±0.08 | ±0.18 | ±0.04 | ±0.05 |
| **MOMENTbase** | 0.903 | 1.220 | 0.755 | 1.104 | 0.910 | 1.113 | 0.946 | 1.218 | 0.906 | 1.251 | 0.776 | 1.203 | 0.428 | 0.415 |
| + HeadOnly | 0.588 | 0.659 | 0.569 | 0.751 | 0.851 | 0.985 | 0.666 | 0.714 | 0.716 | 0.867 | 0.550 | 0.792 | 0.297 | 0.264 |
|  | ±0.08 | ±0.13 | ±0.06 | ±0.13 | ±0.03 | ±0.04 | ±0.08 | ±0.12 | ±0.07 | ±0.13 | ±0.09 | ±0.18 | ±0.05 | ±0.05 |
| + Full FT | 0.470 | 0.465 | 0.481 | 0.599 | 0.817 | 0.923 | 0.629 | 0.631 | 0.553 | 0.566 | 0.559 | 0.809 | 0.327 | 0.294 |
|  | ±0.14 | ±0.22 | ±0.13 | ±0.26 | ±0.05 | ±0.09 | ±0.10 | ±0.14 | ±0.16 | ±0.23 | ±0.10 | ±0.19 | ±0.06 | ±0.07 |
| + LoRA | 0.541 | 0.594 | 0.538 | 0.691 | 0.843 | 0.971 | 0.625 | 0.654 | 0.677 | 0.797 | 0.547 | 0.788 | 0.294 | 0.259 |
|  | ±0.09 | ±0.13 | ±0.07 | ±0.14 | ±0.03 | ±0.05 | ±0.08 | ±0.12 | ±0.08 | ±0.13 | ±0.09 | ±0.18 | ±0.05 | ±0.05 |
| + FourierFT | 0.550 | 0.606 | 0.544 | 0.704 | 0.844 | 0.974 | 0.634 | 0.667 | 0.685 | 0.811 | 0.548 | 0.788 | 0.294 | 0.259 |
|  | ±0.08 | ±0.13 | ±0.06 | ±0.14 | ±0.03 | ±0.05 | ±0.08 | ±0.12 | ±0.08 | ±0.13 | ±0.09 | ±0.18 | ±0.05 | ±0.05 |
| + ChannelMixing | 0.631 | 0.734 | 0.530 | 0.673 | 0.856 | 1.001 | 0.613 | 0.649 | 0.667 | 0.761 | 0.527 | 0.721 | 0.266 | 0.215 |
|  | ±0.11 | ±0.19 | ±0.07 | ±0.15 | ±0.03 | ±0.05 | ±0.10 | ±0.15 | ±0.09 | ±0.15 | ±0.08 | ±0.18 | ±0.05 | ±0.05 |
| + *Time-PEFT* | **0.385** | **0.385** | **0.405** | **0.469** | **0.807** | **0.916** | **0.452** | **0.428** | **0.492** | **0.494** | **0.459** | **0.682** | **0.248** | **0.186** |
|  | ±0.14 | ±0.20 | ±0.10 | ±0.17 | ±0.05 | ±0.07 | ±0.11 | ±0.15 | ±0.15 | ±0.20 | ±0.08 | ±0.18 | ±0.05 | ±0.05 |
| **UniTS** | 0.897 | 1.203 | 0.752 | 1.090 | 0.904 | 1.091 | 0.942 | 1.203 | 0.902 | 1.239 | 0.773 | 1.194 | 0.427 | 0.411 |
| + HeadOnly | 0.500 | 0.532 | 0.456 | 0.546 | 0.814 | 0.922 | 0.531 | 0.509 | 0.609 | 0.651 | 0.505 | 0.704 | 0.267 | 0.225 |
|  | ±0.12 | ±0.17 | ±0.09 | ±0.18 | ±0.05 | ±0.08 | ±0.11 | ±0.16 | ±0.11 | ±0.17 | ±0.11 | ±0.22 | ±0.05 | ±0.06 |
| + Full FT | **0.200** | **0.161** | 0.219 | 0.199 | 0.797 | 1.118 | 0.262 | 0.202 | **0.280** | 0.323 | 0.432 | 0.702 | 0.203 | 0.137 |
|  | ±0.13 | ±0.15 | ±0.19 | ±0.23 | ±0.11 | ±0.20 | ±0.14 | ±0.16 | ±0.17 | ±0.26 | ±0.10 | ±0.26 | ±0.06 | ±0.07 |
| + LoRA | 0.239 | 0.212 | 0.241 | 0.217 | 0.766 | 0.887 | 0.255 | 0.203 | 0.301 | 0.298 | 0.414 | 0.609 | 0.197 | 0.121 |
|  | ±0.13 | ±0.17 | ±0.15 | ±0.20 | ±0.08 | ±0.11 | ±0.12 | ±0.14 | ±0.18 | ±0.24 | ±0.10 | ±0.22 | ±0.05 | ±0.05 |
| + FourierFT | 0.499 | 0.530 | 0.452 | 0.542 | 0.814 | 0.922 | 0.534 | 0.515 | 0.607 | 0.648 | 0.475 | 0.655 | 0.272 | 0.232 |
|  | ±0.11 | ±0.17 | ±0.10 | ±0.18 | ±0.05 | ±0.08 | ±0.11 | ±0.15 | ±0.12 | ±0.17 | ±0.08 | ±0.18 | ±0.05 | ±0.06 |
| + ChannelMixing | 0.533 | 0.574 | 0.530 | 0.703 | 0.832 | 0.956 | 0.657 | 0.719 | 0.660 | 0.755 | 0.581 | 0.947 | 0.270 | 0.225 |
|  | ±0.13 | ±0.20 | ±0.14 | ±0.31 | ±0.04 | ±0.06 | ±0.13 | ±0.21 | ±0.16 | ±0.27 | ±0.25 | ±0.66 | ±0.08 | ±0.09 |
| + *Time-PEFT* | 0.207 | 0.177 | **0.173** | **0.137** | **0.761** | **0.869** | **0.223** | **0.171** | 0.290 | **0.275** | **0.399** | **0.548** | **0.195** | **0.119** |
|  | ±0.13 | ±0.16 | ±0.09 | ±0.10 | ±0.08 | ±0.12 | ±0.11 | ±0.12 | ±0.17 | ±0.22 | ±0.09 | ±0.19 | ±0.05 | ±0.05 |
| **Chronossmall** | 0.900 | 1.397 | 0.776 | 1.359 | 1.074 | 1.798 | 0.953 | 1.409 | 0.895 | 1.306 | 0.794 | 1.791 | 0.439 | 0.643 |
| + HeadOnly | 0.645 | 0.738 | 0.655 | 0.800 | 0.849 | 0.969 | 0.649 | 0.712 | 0.759 | 0.858 | 0.649 | 0.884 | 0.390 | 0.320 |
|  | ±0.07 | ±0.13 | ±0.07 | ±0.15 | ±0.03 | ±0.04 | ±0.10 | ±0.16 | ±0.06 | ±0.10 | ±0.03 | ±0.08 | ±0.01 | ±0.03 |
| + Full FT | 0.292 | 0.370 | 0.401 | 0.522 | **0.764** | 0.874 | 0.442 | 0.467 | 0.439 | 0.532 | 0.710 | 1.240 | 0.483 | 0.541 |
|  | ±0.16 | ±0.30 | ±0.17 | ±0.29 | ±0.07 | ±0.10 | ±0.15 | ±0.21 | ±0.20 | ±0.31 | ±0.15 | ±0.48 | ±0.10 | ±0.21 |
| + LoRA | 0.489 | 0.544 | 0.597 | 0.760 | 0.831 | 0.960 | 0.578 | 0.653 | 0.648 | 0.736 | 0.753 | 1.210 | 0.506 | 0.604 |
|  | ±0.15 | ±0.26 | ±0.12 | ±0.25 | ±0.03 | ±0.05 | ±0.14 | ±0.23 | ±0.12 | ±0.20 | ±0.13 | ±0.36 | ±0.09 | ±0.23 |
| + FourierFT | 0.645 | 0.737 | 0.656 | 0.802 | 0.849 | 0.969 | 0.649 | 0.712 | 0.759 | 0.859 | 0.649 | 0.885 | 0.390 | 0.319 |
|  | ±0.07 | ±0.13 | ±0.07 | ±0.15 | ±0.03 | ±0.04 | ±0.10 | ±0.16 | ±0.06 | ±0.10 | ±0.03 | ±0.08 | ±0.01 | ±0.02 |
| + ChannelMixing | 0.675 | 0.819 | 0.606 | 0.824 | 0.878 | 1.069 | 0.661 | 0.727 | 0.750 | 0.936 | 0.629 | 0.904 | 0.357 | 0.321 |
|  | ±0.13 | ±0.24 | ±0.10 | ±0.23 | ±0.04 | ±0.12 | ±0.14 | ±0.23 | ±0.14 | ±0.28 | ±0.04 | ±0.24 | ±0.05 | ±0.07 |
| + *Time-PEFT* | **0.284** | **0.286** | **0.394** | **0.440** | 0.775 | **0.864** | **0.407** | **0.377** | **0.412** | **0.416** | **0.533** | **0.715** | **0.316** | **0.232** |
|  | ±0.14 | ±0.21 | ±0.14 | ±0.22 | ±0.06 | ±0.08 | ±0.14 | ±0.19 | ±0.16 | ±0.23 | ±0.03 | ±0.08 | ±0.02 | ±0.03 |
| **TTMr2** | 0.910 | 1.265 | 0.789 | 1.212 | 0.948 | 1.274 | 0.862 | 1.108 | 0.841 | 1.092 | 0.694 | 1.125 | 0.382 | 0.394 |
| + HeadOnly | 0.671 | 0.798 | 0.586 | 0.737 | 0.852 | 0.978 | 0.638 | 0.650 | 0.743 | 0.844 | 0.512 | 0.685 | 0.277 | 0.236 |
|  | ±0.04 | ±0.05 | ±0.05 | ±0.10 | ±0.02 | ±0.03 | ±0.08 | ±0.11 | ±0.04 | ±0.06 | ±0.07 | ±0.13 | ±0.04 | ±0.03 |
| + Full FT | 0.509 | 0.538 | 0.443 | 0.501 | 0.830 | **0.940** | 0.490 | 0.449 | 0.610 | 0.634 | 0.422 | **0.513** | 0.226 | 0.160 |
|  | ±0.11 | ±0.17 | ±0.10 | ±0.18 | ±0.03 | ±0.05 | ±0.10 | ±0.14 | ±0.11 | ±0.16 | ±0.08 | ±0.14 | ±0.04 | ±0.04 |
| + LoRA | 0.566 | 0.625 | 0.489 | 0.570 | 0.841 | 0.954 | 0.525 | 0.493 | 0.663 | 0.713 | 0.440 | 0.544 | 0.237 | 0.179 |
|  | ±0.08 | ±0.13 | ±0.09 | ±0.17 | ±0.03 | ±0.04 | ±0.11 | ±0.14 | ±0.08 | ±0.12 | ±0.08 | ±0.15 | ±0.04 | ±0.05 |
| + FourierFT | 0.671 | 0.798 | 0.586 | 0.737 | 0.852 | 0.978 | 0.638 | 0.650 | 0.743 | 0.844 | 0.512 | 0.685 | 0.277 | 0.236 |
|  | ±0.04 | ±0.05 | ±0.05 | ±0.10 | ±0.02 | ±0.03 | ±0.08 | ±0.11 | ±0.04 | ±0.06 | ±0.07 | ±0.13 | ±0.04 | ±0.03 |
| + ChannelMixing | 0.682 | 0.804 | 0.614 | 0.729 | 0.853 | 0.969 | 0.619 | 0.621 | 0.739 | 0.817 | 0.535 | 0.660 | 0.300 | 0.218 |
|  | ±0.03 | ±0.06 | ±0.05 | ±0.10 | ±0.02 | ±0.03 | ±0.09 | ±0.14 | ±0.04 | ±0.07 | ±0.03 | ±0.07 | ±0.01 | ±0.02 |
| + *Time-PEFT* | **0.438** | **0.472** | **0.357** | **0.372** | **0.827** | 0.958 | **0.383** | **0.334** | **0.521** | **0.550** | **0.403** | 0.533 | **0.211** | **0.140** |
|  | ±0.12 | ±0.18 | ±0.12 | ±0.19 | ±0.04 | ±0.06 | ±0.11 | ±0.14 | ±0.15 | ±0.22 | ±0.08 | ±0.14 | ±0.04 | ±0.05 |

*Table 7.* Average MAE and MSE for long-term forecasting across forecast horizons {96, 192, 336} on complex datasets for end-to-end models, MOMENT$_{small}$, and MOMENT$_{base}$. Note that the zero-shot results in each TSFM are colored in gray.

| Method | Horizon | Lorenz | | CellCycle | | DPendulum | | Hopfield | | LCoupled | | ECGCA115 | | ECGCA515 | |
|---|---|---|---|---|---|---|---|---|---|---|---|---|---|---|---|
| | | MAE | MSE | MAE | MSE | MAE | MSE | MAE | MSE | MAE | MSE | MAE | MSE | MAE | MSE |
| PatchTST | 96 | 0.413 | 0.390 | 0.396 | 0.425 | 0.807 | 0.912 | 0.462 | 0.430 | 0.536 | 0.556 | 0.379 | 0.480 | 0.218 | 0.180 |
| | 192 | 0.529 | 0.572 | 0.489 | 0.617 | 0.851 | 0.982 | 0.600 | 0.613 | 0.653 | 0.734 | 0.484 | 0.707 | 0.275 | 0.239 |
| | 336 | 0.623 | 0.715 | 0.580 | 0.792 | 0.873 | 1.017 | 0.698 | 0.747 | 0.741 | 0.870 | 0.589 | 0.911 | 0.340 | 0.303 |
| iTransformer | 96 | 0.458 | 0.446 | 0.313 | 0.270 | 0.791 | 0.882 | 0.345 | 0.266 | 0.574 | 0.599 | 0.344 | 0.386 | 0.202 | 0.175 |
| | 192 | 0.603 | 0.677 | 0.455 | 0.555 | 0.850 | 0.972 | 0.499 | 0.485 | 0.716 | 0.823 | 0.460 | 0.647 | 0.264 | 0.234 |
| | 336 | 0.678 | 0.799 | 0.567 | 0.771 | 0.876 | 1.014 | 0.634 | 0.672 | 0.794 | 0.945 | 0.572 | 0.866 | 0.331 | 0.298 |
| MOMENT$_{small}$ | 96 | 0.900 | 1.210 | 0.750 | 1.085 | 0.911 | 1.123 | 0.947 | 1.228 | 0.888 | 1.205 | 0.716 | 1.062 | 0.390 | 0.372 |
| | 192 | 0.914 | 1.247 | 0.752 | 1.106 | 0.916 | 1.136 | 0.954 | 1.240 | 0.914 | 1.275 | 0.772 | 1.204 | 0.428 | 0.419 |
| | 336 | 0.915 | 1.256 | 0.775 | 1.167 | 0.919 | 1.143 | 0.950 | 1.234 | 0.928 | 1.310 | 0.842 | 1.355 | 0.469 | 0.461 |
| + HeadOnly | 96 | 0.532 | 0.554 | 0.518 | 0.639 | 0.822 | 0.936 | 0.594 | 0.598 | 0.653 | 0.751 | 0.453 | 0.595 | 0.239 | 0.208 |
| | 192 | 0.628 | 0.717 | 0.585 | 0.791 | 0.860 | 1.001 | 0.702 | 0.758 | 0.749 | 0.924 | 0.559 | 0.808 | 0.301 | 0.267 |
| | 336 | 0.687 | 0.819 | 0.641 | 0.921 | 0.879 | 1.034 | 0.762 | 0.844 | 0.799 | 1.015 | 0.671 | 1.025 | 0.368 | 0.334 |
| + Full FT | 96 | 0.242 | 0.139 | 0.266 | 0.212 | 0.743 | 0.805 | 0.489 | 0.419 | 0.329 | 0.242 | 0.420 | 0.524 | 0.242 | 0.201 |
| | 192 | 0.448 | 0.425 | 0.457 | 0.553 | 0.816 | 0.920 | 0.643 | 0.651 | 0.565 | 0.571 | 0.544 | 0.783 | 0.302 | 0.264 |
| | 336 | 0.611 | 0.679 | 0.601 | 0.826 | 0.858 | 0.993 | 0.730 | 0.776 | 0.705 | 0.790 | 0.658 | 0.998 | 0.396 | 0.367 |
| + LoRA | 96 | 0.476 | 0.476 | 0.484 | 0.570 | 0.811 | 0.917 | 0.549 | 0.534 | 0.607 | 0.675 | 0.452 | 0.592 | 0.236 | 0.205 |
| | 192 | 0.588 | 0.656 | 0.563 | 0.742 | 0.854 | 0.989 | 0.666 | 0.705 | 0.720 | 0.868 | 0.557 | 0.805 | 0.300 | 0.264 |
| | 336 | 0.658 | 0.773 | 0.626 | 0.885 | 0.875 | 1.026 | 0.736 | 0.803 | 0.779 | 0.971 | 0.669 | 1.022 | 0.364 | 0.330 |
| + FourierFT | 96 | 0.483 | 0.485 | 0.490 | 0.581 | 0.812 | 0.919 | 0.556 | 0.544 | 0.613 | 0.685 | 0.451 | 0.591 | 0.236 | 0.205 |
| | 192 | 0.594 | 0.665 | 0.566 | 0.750 | 0.855 | 0.991 | 0.672 | 0.713 | 0.724 | 0.876 | 0.557 | 0.805 | 0.298 | 0.263 |
| | 336 | 0.662 | 0.779 | 0.629 | 0.891 | 0.875 | 1.027 | 0.740 | 0.810 | 0.782 | 0.977 | 0.669 | 1.022 | 0.364 | 0.330 |
| + ChannelMixing | 96 | 0.555 | 0.588 | 0.459 | 0.512 | 0.813 | 0.927 | 0.485 | 0.455 | 0.553 | 0.562 | 0.454 | 0.547 | 0.226 | 0.176 |
| | 192 | 0.606 | 0.681 | 0.556 | 0.719 | 0.858 | 0.998 | 0.637 | 0.676 | 0.688 | 0.793 | 0.536 | 0.732 | 0.272 | 0.221 |
| | 336 | 0.673 | 0.793 | 0.600 | 0.824 | 0.880 | 1.038 | 0.710 | 0.786 | 0.764 | 0.932 | 0.650 | 0.969 | 0.343 | 0.296 |
| + *Time-PEFT* | 96 | 0.246 | 0.179 | 0.314 | 0.290 | 0.756 | 0.829 | 0.347 | 0.280 | 0.344 | 0.280 | 0.356 | 0.454 | 0.197 | 0.130 |
| | 192 | 0.445 | 0.462 | 0.438 | 0.531 | 0.826 | 0.938 | 0.491 | 0.475 | 0.554 | 0.573 | 0.461 | 0.669 | 0.250 | 0.187 |
| | 336 | 0.569 | 0.644 | 0.539 | 0.719 | 0.857 | 0.988 | 0.606 | 0.628 | 0.675 | 0.749 | 0.559 | 0.886 | 0.307 | 0.254 |
| MOMENT$_{base}$ | 96 | 0.893 | 1.193 | 0.745 | 1.069 | 0.906 | 1.101 | 0.942 | 1.211 | 0.884 | 1.193 | 0.716 | 1.059 | 0.389 | 0.370 |
| | 192 | 0.907 | 1.229 | 0.748 | 1.092 | 0.910 | 1.114 | 0.950 | 1.223 | 0.910 | 1.262 | 0.770 | 1.199 | 0.427 | 0.417 |
| | 336 | 0.909 | 1.239 | 0.771 | 1.152 | 0.913 | 1.122 | 0.946 | 1.219 | 0.924 | 1.298 | 0.841 | 1.350 | 0.468 | 0.459 |
| + HeadOnly | 96 | 0.492 | 0.504 | 0.497 | 0.594 | 0.817 | 0.928 | 0.567 | 0.570 | 0.625 | 0.709 | 0.441 | 0.577 | 0.234 | 0.204 |
| | 192 | 0.602 | 0.681 | 0.573 | 0.762 | 0.858 | 0.997 | 0.682 | 0.738 | 0.733 | 0.897 | 0.548 | 0.792 | 0.295 | 0.260 |
| | 336 | 0.669 | 0.792 | 0.636 | 0.899 | 0.877 | 1.031 | 0.749 | 0.832 | 0.789 | 0.996 | 0.660 | 1.008 | 0.361 | 0.327 |
| + Full FT | 96 | 0.292 | 0.193 | 0.315 | 0.272 | 0.747 | 0.807 | 0.505 | 0.446 | 0.352 | 0.274 | 0.444 | 0.585 | 0.277 | 0.235 |
| | 192 | 0.479 | 0.476 | 0.498 | 0.629 | 0.831 | 0.943 | 0.653 | 0.668 | 0.573 | 0.584 | 0.564 | 0.817 | 0.308 | 0.273 |
| | 336 | 0.639 | 0.727 | 0.630 | 0.895 | 0.874 | 1.019 | 0.729 | 0.778 | 0.733 | 0.839 | 0.668 | 1.025 | 0.397 | 0.374 |
| + LoRA | 96 | 0.430 | 0.423 | 0.455 | 0.514 | 0.804 | 0.907 | 0.517 | 0.499 | 0.569 | 0.619 | 0.440 | 0.573 | 0.232 | 0.199 |
| | 192 | 0.556 | 0.616 | 0.544 | 0.703 | 0.851 | 0.984 | 0.640 | 0.676 | 0.697 | 0.829 | 0.546 | 0.788 | 0.292 | 0.255 |
| | 336 | 0.636 | 0.742 | 0.616 | 0.855 | 0.873 | 1.023 | 0.718 | 0.786 | 0.764 | 0.943 | 0.657 | 1.003 | 0.358 | 0.323 |
| + FourierFT | 96 | 0.441 | 0.437 | 0.463 | 0.531 | 0.807 | 0.911 | 0.527 | 0.514 | 0.580 | 0.636 | 0.440 | 0.573 | 0.231 | 0.199 |
| | 192 | 0.565 | 0.628 | 0.550 | 0.716 | 0.852 | 0.986 | 0.650 | 0.692 | 0.704 | 0.844 | 0.546 | 0.788 | 0.292 | 0.256 |
| | 336 | 0.643 | 0.752 | 0.620 | 0.864 | 0.874 | 1.024 | 0.725 | 0.796 | 0.769 | 0.953 | 0.658 | 1.004 | 0.358 | 0.323 |
| + ChannelMixing | 96 | 0.494 | 0.498 | 0.451 | 0.490 | 0.820 | 0.942 | 0.511 | 0.490 | 0.561 | 0.597 | 0.446 | 0.529 | 0.213 | 0.162 |
| | 192 | 0.717 | 0.894 | 0.536 | 0.693 | 0.866 | 1.014 | 0.608 | 0.647 | 0.672 | 0.763 | 0.509 | 0.690 | 0.259 | 0.206 |
| | 336 | 0.681 | 0.808 | 0.604 | 0.837 | 0.883 | 1.047 | 0.721 | 0.810 | 0.766 | 0.922 | 0.627 | 0.945 | 0.327 | 0.277 |
| + *Time-PEFT* | 96 | 0.206 | 0.134 | 0.284 | 0.249 | 0.743 | 0.817 | 0.312 | 0.244 | 0.299 | 0.228 | 0.355 | 0.462 | 0.193 | 0.125 |
| | 192 | 0.403 | 0.408 | 0.410 | 0.484 | 0.821 | 0.938 | 0.461 | 0.440 | 0.522 | 0.532 | 0.463 | 0.683 | 0.247 | 0.181 |
| | 336 | 0.545 | 0.614 | 0.521 | 0.675 | 0.856 | 0.994 | 0.583 | 0.599 | 0.655 | 0.723 | 0.559 | 0.900 | 0.304 | 0.250 |

*Table 8.* Average MAE and MSE for long-term forecasting across forecast horizons {96, 192, 336} on complex datasets for UniTS, encoder-based Chronos$_{small}$, and encoder-based TTM$_{r2}$. Note that the zero-shot results in each TSFM are colored in gray.

| Method | Horizon | Lorenz MAE | Lorenz MSE | CellCycle MAE | CellCycle MSE | DPendulum MAE | DPendulum MSE | Hopfield MAE | Hopfield MSE | LCoupled MAE | LCoupled MSE | ECGCA115 MAE | ECGCA115 MSE | ECGCA515 MAE | ECGCA515 MSE |
|---|---|---|---|---|---|---|---|---|---|---|---|---|---|---|---|
| UniTS | 96 | 0.887 | 1.176 | 0.743 | 1.057 | 0.900 | 1.080 | 0.939 | 1.197 | 0.879 | 1.181 | 0.713 | 1.050 | 0.388 | 0.366 |
| | 192 | 0.901 | 1.212 | 0.745 | 1.078 | 0.905 | 1.093 | 0.946 | 1.208 | 0.906 | 1.250 | 0.767 | 1.189 | 0.426 | 0.412 |
| | 336 | 0.903 | 1.222 | 0.768 | 1.137 | 0.908 | 1.101 | 0.942 | 1.203 | 0.920 | 1.286 | 0.839 | 1.342 | 0.466 | 0.455 |
| + HeadOnly | 96 | 0.348 | 0.303 | 0.341 | 0.330 | 0.750 | 0.822 | 0.390 | 0.306 | 0.461 | 0.429 | 0.388 | 0.463 | 0.205 | 0.162 |
| | 192 | 0.528 | 0.571 | 0.460 | 0.552 | 0.828 | 0.942 | 0.556 | 0.542 | 0.635 | 0.688 | 0.480 | 0.666 | 0.265 | 0.220 |
| | 336 | 0.625 | 0.723 | 0.566 | 0.756 | 0.863 | 1.001 | 0.648 | 0.679 | 0.730 | 0.835 | 0.646 | 0.984 | 0.331 | 0.295 |
| + Full FT | 96 | 0.077 | 0.023 | 0.069 | 0.026 | 0.645 | 0.842 | 0.100 | 0.042 | 0.101 | 0.061 | 0.307 | 0.376 | 0.136 | 0.059 |
| | 192 | 0.161 | 0.096 | 0.158 | 0.113 | 0.833 | 1.191 | 0.262 | 0.167 | 0.239 | 0.241 | 0.438 | 0.751 | 0.195 | 0.120 |
| | 336 | 0.363 | 0.363 | 0.431 | 0.456 | 0.912 | 1.320 | 0.425 | 0.398 | 0.501 | 0.669 | 0.553 | 0.979 | 0.280 | 0.230 |
| + LoRA | 96 | 0.104 | 0.041 | 0.094 | 0.034 | 0.656 | 0.731 | 0.129 | 0.055 | 0.119 | 0.060 | 0.292 | 0.335 | 0.140 | 0.060 |
| | 192 | 0.205 | 0.150 | 0.182 | 0.131 | 0.795 | 0.932 | 0.226 | 0.160 | 0.245 | 0.202 | 0.416 | 0.624 | 0.193 | 0.111 |
| | 336 | 0.409 | 0.446 | 0.446 | 0.487 | 0.848 | 0.998 | 0.410 | 0.395 | 0.538 | 0.631 | 0.535 | 0.867 | 0.257 | 0.191 |
| + FourierFT | 96 | 0.355 | 0.312 | 0.335 | 0.318 | 0.749 | 0.819 | 0.403 | 0.328 | 0.451 | 0.417 | 0.372 | 0.436 | 0.210 | 0.166 |
| | 192 | 0.516 | 0.556 | 0.454 | 0.543 | 0.829 | 0.946 | 0.532 | 0.513 | 0.641 | 0.694 | 0.480 | 0.665 | 0.265 | 0.224 |
| | 336 | 0.625 | 0.723 | 0.567 | 0.765 | 0.863 | 1.000 | 0.667 | 0.705 | 0.729 | 0.831 | 0.575 | 0.864 | 0.340 | 0.306 |
| + ChannelMixing | 96 | 0.375 | 0.334 | 0.381 | 0.381 | 0.780 | 0.874 | 0.513 | 0.507 | 0.467 | 0.434 | 0.373 | 0.442 | 0.197 | 0.146 |
| | 192 | 0.546 | 0.591 | 0.500 | 0.617 | 0.846 | 0.983 | 0.666 | 0.719 | 0.689 | 0.781 | 0.467 | 0.658 | 0.254 | 0.204 |
| | 336 | 0.678 | 0.797 | 0.709 | 1.110 | 0.869 | 1.011 | 0.792 | 0.932 | 0.826 | 1.050 | 0.904 | 1.742 | 0.359 | 0.324 |
| + *Time-PEFT* | 96 | 0.079 | 0.024 | 0.071 | 0.025 | 0.649 | 0.705 | 0.101 | 0.039 | 0.102 | 0.045 | 0.284 | 0.318 | 0.139 | 0.059 |
| | 192 | 0.165 | 0.108 | 0.160 | 0.118 | 0.788 | 0.906 | 0.201 | 0.136 | 0.250 | 0.201 | 0.400 | 0.544 | 0.192 | 0.110 |
| | 336 | 0.379 | 0.398 | 0.287 | 0.268 | 0.846 | 0.996 | 0.368 | 0.337 | 0.518 | 0.579 | 0.512 | 0.781 | 0.254 | 0.188 |
| Chronos$_{small}$ | 96 | 0.841 | 1.294 | 0.740 | 1.305 | 1.048 | 1.752 | 0.928 | 1.397 | 0.842 | 1.213 | 0.674 | 1.687 | 0.378 | 0.653 |
| | 192 | 0.909 | 1.411 | 0.786 | 1.386 | 1.086 | 1.836 | 0.963 | 1.425 | 0.909 | 1.329 | 0.788 | 1.721 | 0.436 | 0.614 |
| | 336 | 0.950 | 1.486 | 0.802 | 1.385 | 1.087 | 1.808 | 0.967 | 1.404 | 0.935 | 1.376 | 0.921 | 1.964 | 0.504 | 0.661 |
| + HeadOnly | 96 | 0.552 | 0.565 | 0.565 | 0.600 | 0.816 | 0.914 | 0.519 | 0.505 | 0.686 | 0.724 | 0.603 | 0.772 | 0.375 | 0.285 |
| | 192 | 0.659 | 0.761 | 0.669 | 0.829 | 0.856 | 0.980 | 0.665 | 0.729 | 0.771 | 0.877 | 0.664 | 0.919 | 0.396 | 0.331 |
| | 336 | 0.725 | 0.887 | 0.731 | 0.972 | 0.877 | 1.014 | 0.763 | 0.903 | 0.820 | 0.973 | 0.679 | 0.961 | 0.399 | 0.343 |
| + Full FT | 96 | 0.120 | 0.057 | 0.196 | 0.158 | 0.670 | 0.745 | 0.246 | 0.190 | 0.192 | 0.140 | 0.541 | 0.703 | 0.362 | 0.306 |
| | 192 | 0.245 | 0.270 | 0.402 | 0.536 | 0.784 | 0.902 | 0.465 | 0.506 | 0.453 | 0.567 | 0.706 | 1.197 | 0.498 | 0.573 |
| | 336 | 0.510 | 0.782 | 0.605 | 0.871 | 0.836 | 0.976 | 0.615 | 0.706 | 0.672 | 0.888 | 0.883 | 1.819 | 0.588 | 0.744 |
| + LoRA | 96 | 0.301 | 0.208 | 0.432 | 0.423 | 0.786 | 0.890 | 0.400 | 0.365 | 0.485 | 0.466 | 0.592 | 0.767 | 0.396 | 0.341 |
| | 192 | 0.511 | 0.590 | 0.633 | 0.836 | 0.843 | 0.978 | 0.584 | 0.661 | 0.688 | 0.809 | 0.754 | 1.224 | 0.498 | 0.567 |
| | 336 | 0.657 | 0.834 | 0.727 | 1.020 | 0.866 | 1.011 | 0.750 | 0.934 | 0.772 | 0.932 | 0.914 | 1.639 | 0.624 | 0.902 |
| + FourierFT | 96 | 0.551 | 0.563 | 0.566 | 0.601 | 0.815 | 0.913 | 0.520 | 0.507 | 0.685 | 0.722 | 0.604 | 0.774 | 0.376 | 0.286 |
| | 192 | 0.659 | 0.762 | 0.669 | 0.829 | 0.855 | 0.977 | 0.665 | 0.725 | 0.772 | 0.880 | 0.664 | 0.919 | 0.395 | 0.330 |
| | 336 | 0.725 | 0.887 | 0.733 | 0.976 | 0.878 | 1.018 | 0.764 | 0.905 | 0.820 | 0.974 | 0.679 | 0.960 | 0.398 | 0.342 |
| + ChannelMixing | 96 | 0.562 | 0.607 | 0.501 | 0.609 | 0.829 | 0.960 | 0.496 | 0.473 | 0.620 | 0.691 | 0.509 | 0.645 | 0.293 | 0.240 |
| | 192 | 0.696 | 0.857 | 0.622 | 0.857 | 0.889 | 1.091 | 0.687 | 0.772 | 0.781 | 0.998 | 0.647 | 0.940 | 0.368 | 0.336 |
| | 336 | 0.768 | 0.993 | 0.697 | 1.007 | 0.915 | 1.155 | 0.799 | 0.937 | 0.850 | 1.120 | 0.731 | 1.127 | 0.411 | 0.387 |
| + *Time-PEFT* | 96 | 0.136 | 0.061 | 0.221 | 0.156 | 0.701 | 0.759 | 0.242 | 0.160 | 0.206 | 0.131 | 0.495 | 0.624 | 0.297 | 0.192 |
| | 192 | 0.251 | 0.225 | 0.405 | 0.460 | 0.789 | 0.885 | 0.394 | 0.357 | 0.424 | 0.431 | 0.527 | 0.704 | 0.316 | 0.234 |
| | 336 | 0.466 | 0.573 | 0.557 | 0.703 | 0.835 | 0.949 | 0.584 | 0.614 | 0.606 | 0.684 | 0.578 | 0.815 | 0.336 | 0.270 |
| TTM$_{r2}$ | 96 | 0.890 | 1.236 | 0.702 | 1.054 | 0.888 | 1.089 | 0.817 | 1.025 | 0.789 | 1.005 | 0.565 | 0.884 | 0.322 | 0.339 |
| | 192 | 0.899 | 1.231 | 0.787 | 1.144 | 0.920 | 1.163 | 0.865 | 1.087 | 0.847 | 1.084 | 0.683 | 1.076 | 0.378 | 0.390 |
| | 336 | 0.940 | 1.326 | 0.879 | 1.436 | 1.035 | 1.569 | 0.905 | 1.211 | 0.887 | 1.185 | 0.833 | 1.414 | 0.447 | 0.453 |
| + HeadOnly | 96 | 0.628 | 0.738 | 0.517 | 0.604 | 0.826 | 0.937 | 0.540 | 0.510 | 0.688 | 0.762 | 0.427 | 0.527 | 0.229 | 0.194 |
| | 192 | 0.668 | 0.789 | 0.590 | 0.747 | 0.858 | 0.985 | 0.648 | 0.665 | 0.749 | 0.850 | 0.523 | 0.689 | 0.282 | 0.237 |
| | 336 | 0.715 | 0.868 | 0.649 | 0.860 | 0.874 | 1.011 | 0.725 | 0.775 | 0.794 | 0.920 | 0.587 | 0.839 | 0.320 | 0.278 |
| + Full FT | 96 | 0.362 | 0.316 | 0.310 | 0.269 | 0.786 | 0.872 | 0.359 | 0.276 | 0.464 | 0.421 | 0.325 | 0.340 | 0.177 | 0.108 |
| | 192 | 0.545 | 0.586 | 0.458 | 0.530 | 0.842 | 0.957 | 0.501 | 0.459 | 0.643 | 0.678 | 0.427 | 0.508 | 0.229 | 0.156 |
| | 336 | 0.622 | 0.713 | 0.560 | 0.705 | 0.864 | 0.992 | 0.609 | 0.612 | 0.722 | 0.803 | 0.513 | 0.693 | 0.273 | 0.217 |
| + LoRA | 96 | 0.451 | 0.447 | 0.367 | 0.356 | 0.803 | 0.895 | 0.390 | 0.313 | 0.555 | 0.549 | 0.340 | 0.362 | 0.186 | 0.120 |
| | 192 | 0.592 | 0.662 | 0.504 | 0.596 | 0.849 | 0.968 | 0.538 | 0.508 | 0.680 | 0.738 | 0.446 | 0.539 | 0.242 | 0.186 |
| | 336 | 0.655 | 0.765 | 0.595 | 0.757 | 0.869 | 0.999 | 0.647 | 0.659 | 0.756 | 0.852 | 0.534 | 0.731 | 0.285 | 0.233 |
| + FourierFT | 96 | 0.628 | 0.738 | 0.517 | 0.604 | 0.826 | 0.937 | 0.540 | 0.509 | 0.688 | 0.762 | 0.427 | 0.527 | 0.229 | 0.194 |
| | 192 | 0.668 | 0.789 | 0.590 | 0.747 | 0.858 | 0.985 | 0.648 | 0.665 | 0.749 | 0.850 | 0.523 | 0.690 | 0.282 | 0.237 |
| | 336 | 0.715 | 0.868 | 0.649 | 0.860 | 0.874 | 1.011 | 0.725 | 0.775 | 0.794 | 0.920 | 0.587 | 0.839 | 0.319 | 0.278 |
| + ChannelMixing | 96 | 0.642 | 0.731 | 0.551 | 0.601 | 0.829 | 0.930 | 0.502 | 0.450 | 0.682 | 0.723 | 0.502 | 0.573 | 0.289 | 0.195 |
| | 192 | 0.688 | 0.817 | 0.621 | 0.743 | 0.859 | 0.977 | 0.631 | 0.638 | 0.748 | 0.834 | 0.539 | 0.667 | 0.300 | 0.216 |
| | 336 | 0.717 | 0.863 | 0.671 | 0.843 | 0.873 | 0.999 | 0.725 | 0.775 | 0.787 | 0.895 | 0.563 | 0.741 | 0.310 | 0.244 |
| + *Time-PEFT* | 96 | 0.291 | 0.262 | 0.202 | 0.130 | 0.769 | 0.873 | 0.253 | 0.169 | 0.322 | 0.256 | 0.307 | 0.375 | 0.163 | 0.089 |
| | 192 | 0.450 | 0.484 | 0.369 | 0.383 | 0.841 | 0.979 | 0.383 | 0.333 | 0.555 | 0.601 | 0.399 | 0.503 | 0.209 | 0.132 |
| | 336 | 0.573 | 0.668 | 0.501 | 0.602 | 0.870 | 1.023 | 0.512 | 0.502 | 0.686 | 0.794 | 0.503 | 0.720 | 0.262 | 0.199 |

*Table 9.* Average MAE and MSE for long-term forecasting across forecast horizons {96, 192, 336} on standard datasets for end-to-end models, MOMENT_small, and MOMENT_base, UniTS, encoder-based Chronos_small, and encoder-based TTM_r2. Within each TSFM block, the best and second-best fine-tuning results are shown in **bold** and underlined, respectively; gray rows denote zero-shot results.

| Method | ETTh1 | | ETTh2 | | ETTm1 | | ETTm2 | | Weather | | Exchange | |
|---|---|---|---|---|---|---|---|---|---|---|---|---|
| | **MAE** | **MSE** | **MAE** | **MSE** | **MAE** | **MSE** | **MAE** | **MSE** | **MAE** | **MSE** | **MAE** | **MSE** |
| PatchTST | 0.427 | 0.435 | 0.391 | 0.362 | 0.387 | 0.366 | 0.304 | 0.242 | 0.261 | 0.231 | 0.312 | 0.204 |
| iTransformer | 0.485 | 0.521 | 0.407 | 0.387 | 0.402 | 0.395 | 0.312 | 0.252 | 0.261 | 0.231 | 0.311 | 0.203 |
| **MOMENT_small** | 0.579 | 0.747 | 0.428 | 0.420 | 0.563 | 0.733 | 0.342 | 0.289 | 0.307 | 0.269 | 0.360 | 0.255 |
| + HeadOnly | 0.427 | 0.425 | 0.388 | 0.353 | 0.381 | 0.353 | 0.304 | 0.241 | 0.262 | 0.231 | 0.308 | 0.199 |
| + Full FT | 0.440 | 0.437 | 0.399 | 0.377 | 0.418 | 0.411 | 0.329 | 0.264 | 0.270 | 0.238 | 0.339 | 0.240 |
| + LoRA | **0.426** | 0.424 | **0.388** | 0.353 | 0.382 | 0.354 | 0.304 | **0.241** | 0.262 | 0.231 | **0.307** | **0.198** |
| + FourierFT | 0.426 | 0.424 | 0.388 | **0.353** | 0.381 | 0.353 | **0.304** | 0.241 | 0.262 | 0.231 | 0.308 | 0.200 |
| + ChannelMixing | 0.446 | 0.458 | 0.411 | 0.393 | 0.387 | 0.359 | 0.315 | 0.258 | 0.275 | 0.239 | 0.359 | 0.257 |
| *+ Time-PEFT* | 0.437 | **0.424** | 0.398 | 0.378 | **0.380** | **0.352** | 0.317 | 0.266 | **0.255** | **0.215** | 0.320 | 0.214 |
| **MOMENT_base** | 0.574 | 0.731 | 0.426 | 0.416 | 0.559 | 0.720 | 0.340 | 0.287 | 0.306 | 0.267 | 0.360 | 0.254 |
| + HeadOnly | 0.430 | 0.431 | **0.391** | **0.358** | **0.380** | **0.353** | **0.305** | **0.242** | 0.261 | 0.229 | 0.309 | 0.201 |
| + Full FT | 0.445 | 0.450 | 0.418 | 0.404 | 0.435 | 0.448 | 0.323 | 0.260 | 0.268 | 0.237 | 0.329 | 0.229 |
| + LoRA | 0.429 | 0.430 | 0.392 | 0.359 | 0.381 | 0.353 | 0.305 | 0.242 | 0.262 | 0.230 | 0.306 | **0.196** |
| + FourierFT | **0.429** | **0.430** | 0.393 | 0.359 | 0.381 | 0.354 | 0.305 | 0.243 | 0.261 | 0.229 | **0.306** | 0.198 |
| + ChannelMixing | 0.458 | 0.500 | 0.409 | 0.385 | 0.406 | 0.411 | 0.316 | 0.257 | 0.273 | 0.238 | 0.352 | 0.251 |
| *+ Time-PEFT* | 0.436 | 0.432 | 0.396 | 0.371 | 0.382 | 0.355 | 0.315 | 0.261 | **0.254** | **0.213** | 0.318 | 0.209 |
| **UniTS** | 0.572 | 0.717 | 0.424 | 0.413 | 0.558 | 0.711 | 0.338 | 0.284 | 0.305 | 0.265 | 0.360 | 0.254 |
| + HeadOnly | **0.441** | **0.453** | **0.392** | **0.368** | 0.388 | 0.369 | 0.304 | 0.241 | 0.254 | 0.219 | 0.306 | 0.200 |
| + Full FT | 0.499 | 0.590 | 0.424 | 0.441 | 0.419 | 0.457 | 0.335 | 0.280 | 0.266 | 0.229 | 0.355 | 0.266 |
| + LoRA | 0.472 | 0.510 | 0.432 | 0.436 | 0.407 | 0.412 | 0.323 | 0.273 | **0.252** | **0.214** | 0.323 | 0.218 |
| + FourierFT | 0.450 | 0.463 | 0.394 | 0.371 | **0.384** | **0.364** | **0.303** | **0.239** | 0.253 | 0.216 | **0.301** | **0.191** |
| + ChannelMixing | 0.481 | 0.510 | 0.424 | 0.410 | 0.410 | 0.398 | 0.329 | 0.271 | 0.286 | 0.251 | 0.354 | 0.244 |
| *+ Time-PEFT* | 0.455 | 0.467 | 0.412 | 0.409 | 0.418 | 0.419 | 0.317 | 0.265 | 0.258 | 0.216 | 0.331 | 0.229 |
| **Chronos_small** | 0.431 | 0.495 | 0.400 | 0.408 | 0.565 | 0.977 | 0.343 | 0.306 | 0.309 | 0.369 | 0.314 | 0.225 |
| + HeadOnly | 0.679 | 0.793 | 1.108 | 2.221 | **0.657** | **0.760** | **1.060** | **2.114** | 0.624 | 0.702 | 1.346 | 2.883 |
| + Full FT | 0.742 | 0.942 | **1.066** | **2.171** | 0.787 | 1.087 | 1.281 | 3.360 | 0.783 | 1.294 | **1.198** | **2.373** |
| + LoRA | 0.704 | 0.931 | 1.217 | 3.120 | 0.811 | 1.323 | 1.361 | 4.174 | 0.794 | 1.300 | 1.380 | 3.261 |
| + FourierFT | **0.674** | **0.786** | 1.106 | 2.213 | 0.663 | 0.768 | 1.074 | 2.156 | 0.626 | 0.705 | 1.334 | 2.829 |
| + ChannelMixing | 0.734 | 0.905 | 1.360 | 3.156 | 0.726 | 0.860 | 1.133 | 2.182 | 0.550 | 0.593 | 1.425 | 2.989 |
| *+ Time-PEFT* | 0.720 | 0.886 | 1.278 | 2.894 | 0.687 | 0.813 | 1.072 | 2.244 | **0.484** | **0.515** | 1.398 | 3.033 |
| **TTM_r2** | 0.407 | 0.384 | 0.378 | 0.335 | 0.379 | 0.373 | 0.305 | 0.249 | 0.241 | 0.200 | 0.309 | 0.199 |
| + HeadOnly | 0.427 | 0.414 | 0.375 | 0.325 | 0.363 | 0.329 | 0.290 | 0.215 | 0.239 | 0.195 | 0.329 | 0.226 |
| + Full FT | **0.425** | **0.405** | 0.380 | 0.333 | **0.361** | **0.319** | 0.292 | 0.219 | 0.235 | 0.192 | **0.318** | **0.209** |
| + LoRA | 0.429 | 0.412 | 0.378 | 0.330 | 0.362 | 0.324 | 0.292 | 0.218 | **0.235** | **0.191** | 0.320 | 0.213 |
| + FourierFT | 0.428 | 0.414 | **0.374** | **0.324** | 0.363 | 0.329 | **0.290** | **0.215** | 0.239 | 0.195 | 0.329 | 0.225 |
| + ChannelMixing | 0.771 | 0.962 | 1.385 | 3.256 | 0.753 | 0.897 | 1.419 | 3.279 | 0.487 | 0.492 | 1.517 | 3.539 |
| *+ Time-PEFT* | 0.446 | 0.435 | 0.394 | 0.356 | 0.379 | 0.344 | 0.316 | 0.262 | 0.250 | 0.206 | 0.401 | 0.356 |

