# OpenReview forum: "Time-PEFT: Temporal and Multichannel Complexity-Based Fine-Tuning for Time-Series Foundation Models"
_ICML.cc/2026/Conference — ICML 2026 regular_

### Official Review · Reviewer_fAnw · 2026-02-27

**Soundness:** 2
**Presentation:** 3
**Significance:** 3
**Originality:** 2
**Overall Recommendation:** 3
**Confidence:** 5

**Summary:**

The paper study the adaptation of Time-Series Foundation Models for complex, real-world downstream tasks. The paper introduces two data-driven metrics Temporal Complexity and Multichannel Complexity to quantify the difficulty of time-series datasets. Based on these metrics, the authors propose Time-PEFT, a parameter-efficient fine-tuning framework consisting of a Frequency Adapter for top-$k$ filtering and a Channel Adapter for multichannel modeling. Extensive experiments demonstrate that Time-PEFT outperforms standard PEFT baselines on highly complex datasets. While the theoretical motivation from Kolmogorov-Szegő formula and Fano's Inequality is interesting and the paper is well-written, there remain several issues that require further clarification and improvement.

**Compliance With Llm Reviewing Policy:**

Affirmed.

**Final Justification:**

I maintain my weak reject recommendation. The contribution is limited since both core modules are simply combinations of existing components. More importantly, parameter shared projection and explicit cross-channel operations are fundamentally different concepts.

**Key Questions For Authors:**

1. Please provide mathematical or empirical justification that applying Top-$k$ FFT masking on deep latent embeddings naturally aligns with the dominant frequencies of the raw data. Clarify the specific advantages of performing this in the embedding space rather than on the raw time-series.

2. Table 4 shows that combining the Frequency Adapter and LoRA underperforms using LoRA alone. Given this performance drop, please clarify the core motivation for this module and provide evidence that it effectively filters noise.

3. Please explain how the Channel Adapter explicitly captures the directed information flow measured by Multichannel Complexity when its forward pass entirely lacks cross-channel operations.

4. Please provide additional experiments on datasets with a significantly larger number of channels to validate the true scalability of the Channel Adapter.

5. To strengthen the empirical claims, please include fine-tuning evaluations on newer models such as Moirai-2, TimesFM2, and Time-MoE.

6. Please justify the rationale for using different feature dimensions $h_1$ and $h_2$ for the adapters. Additionally, provide an empirical comparison between element-wise addition and concatenation for fusing the filtered and backbone embeddings.

**Limitations:**

The authors should deeply investigate the mathematical implications of applying FFT to latent embeddings to clarify whether this approach can effectively model temporal dependencies. Furthermore, the authors should explore more explicit methods for modeling channel correlations and validate their framework on datasets with a significantly larger number of channels. Finally, the authors should conduct experiments using the most recent time-series foundation models as backbones.

**Strengths And Weaknesses:**

Strengths:
1. The theoretical motivation derived from the Kolmogorov-Szegő formula and Fano's inequality is interesting.
2. The paper is well-written with a clear logical flow making the complex concepts and mathematical definitions easy to follow.
3. The visual illustrations and architecture diagrams are of high quality.
4. Introducing data-driven metrics to quantify temporal and multichannel complexity serves as a practical and valuable tool.

Weaknesses：
1. Selecting top-$k$ frequencies via FFT is standard practice in time-series forecasting like TimesNet, FEDformer, FiLM. This limits the methodological novelty of Frequency Adapter.  The authors need to clarify the essential differences between the proposed Frequency Adapter and these existing methods.
2. Temporal Complexity is computed on raw 1D data, but the Frequency Adapter applies FFT to high-dimensional latent embeddings. It remains unclear whether the frequency structure of raw data naturally aligns with deep latent semantics, lacking both theoretical and empirical justification.
3. As shown in Table 4, adding only the Frequency Adapter actually degrades performance compared to the LoRA-only baseline. The authors need to explain this performance drop.
4. Despite claims of learning channel correlations, the forward pass is strictly channel-independent. It essentially functions as a two-layer MLP. the design does not fully support the claim of capturing explicit channel dependencies.
5. The datasets used to demonstrate multichannel complexity contain only 3 to 6 channels. To truly validate the module's capacity to capture complex inter-variable dependencies, it should be evaluated on standard high-dimensional datasets containing dozens or hundreds of variables.
6. Including newer architectures like Moirai-2, TimesFM2, or Time-MoE is necessary to demonstrate the method's generalizability and relevance in a rapidly advancing research landscape.

[1] TimesNet: Temporal 2D-Variation Modeling for General Time Series Analysis.ICLR 2023 \
[2] FEDformer: Frequency Enhanced Decomposed Transformer for Long-term Series Forecasting. ICML 2022 \
[3] FiLM: Frequency improved Legendre Memory Model for Long-term Time Series Forecasting. NeurIPS 2022 \
[4] Moirai 2.0: When Less Is More for Time Series Forecasting. arXiv 2025\
[5] Time-MoE: Billion-Scale Time Series Foundation Models with Mixture of Experts. ICLR 2025

---

> ### Author Rebuttal · Authors · 2026-03-31
>
> We appreciate the careful and constructive feedback.
>
> >W1. Differences between the frequency adapter and existing methods.
>
> We agree that FFT-based top-k selection itself is not a new technique. We will cite prior works. The novelty of our work lies in the overall proposed method, the proposed indicators, and the performance gains achieved by adapting the model based on those indicators. Therefore, even if the standalone novelty of the frequency adapter may appear somewhat limited, we would greatly appreciate it if the contribution could be assessed in terms of the **complementary roles of the frequency and the channel adapter**.
>
> > [V] {W2.+Q1.} Top-k on embeddings and raw data.
>
> As shown in Table F, when we use the raw time series as the input to the frequency adapter and the backbone embedding as the input, the results are similar. This **empirically suggests that the effect of top-k masking can be realized not only in the raw time-series space but also in the embedding space**. Because TSFMs employ diverse backbones, the projection of masked **raw** time series may become substantially misaligned with the backbone embedding. This concern motivated our design choice.
>
> Table F. Average MSE on MOMENT$_{base}$ with horizon={96,192,336}.
> ||Lorenz|CellCycle|DPendulum|Hopfield|LCoupled|ecgca115|ecgca515|
> |-|-|-|-|-|-|-|-|
> |Raw| 0.44 |0.48|0.92|0.42|0.53|0.68|0.20|
> |Embedding|0.49| 0.53 |0.92|0.48|0.57|0.68|0.20|
>
> > {W3+Q2} Motivation of frequency adapter.
>
> The frequency adapter is **not its standalone effect, but rather complementary roles with the channel adapter**. The degree to which each channel should rely on the backbone embedding versus the filtered embedding may differ across channels in datasets with high temporal complexity. Therefore, if the forecast head is applied immediately after the frequency adapter, it becomes difficult for the head to determine the relative importance of the backbone and filtered embeddings for each channel. In contrast, the channel adapter enables channel-specific consideration of both the filtered and backbone embeddings. Accordingly, even if the frequency adapter alone is not effective, its synergy with the channel adapter allows top-k filtering to function effectively.
>
> > {W4+Q3} Cross-channel operations.
>
> The channel-specific up-projections allow each channel to learn its own essential characteristics. Prior to that, the shared down-projection learns **implicit channel dependency**. The shared projection is encouraged to capture information that spans multiple channels rather than information specific to any single channel. In this way, the **common information extracted across channels is passed to each individual channel**, allowing each channel to make use of information from the others. Since the module does not explicitly have directed information flow, we agree with the reviewer’s point and will revise the paper. Nevertheless, the strong performance gain of LoRA + Channel Adapter in Table 4 empirically shows the effectiveness of the channel adapter.
>
> >[v] {W5+Q4} Hgh-dimensional datasets.
>
> The complex datasets considered in this work are not simply high-dimensional datasets. Our notion of multichannel complexity refers to the extent to which using information from other channels is helpful for future prediction, rather than modeling each channel independently. Therefore, a **high-dimensional dataset does not necessarily have high channel complexity.** In the paper, the Weather dataset (# channels = 21) has low multichannel complexity despite having many channels. Therefore, the evaluation on high-dimensional standard datasets lies **outside the main scope** of this work, which aims to improve fine-tuning for complex datasets.
>
>
> > [V] {W6+Q5} Recent TSFMs.
>
> We will additionally include **Time-MoE**, which is a more recent TSFM. As an initial step, we share partial results in Table G, and we will include the full set of results in the final version.
>
> Table G. MSE on Time-MoE with horizon=96.
> ||Lorenz|CellCycle|DPendulum|Hopfield|LCoupled|ecgca115|ecgca515|
> |-|-|-|-|-|-|-|-|
> |Zeroshot|1.19|0.96|1.05|0.97|1.12|0.90|0.33|
> |LoRA|0.04|0.09|0.83|0.20|0.76|0.51|0.14|
> |Time-PEFT|0.03|0.07|0.86|0.11|0.10|0.43|0.12|
>
> > {Q6} (1) Different h_1 and h_2. (2) Fusion.
>
> (1) We used dimensions of h_1 and h_2 to make it easier to explain the roles of the different components. However, in the actual experiments, we set h_1 and h_2 to be the **same** as the backbone embedding dimension of each TSFM.
>
> (2) In Table H, **concatenation (OURS) performs better** than element-wise addition for fusing the filtered and backbone embeddings by helping the model better identify ambiguous or confusing aspects in the backbone representation.
>
> Table H. Average MSE on MOMOENT$_{base}$ with horizon={96,192,336}.
> ||Lorenz|CellCycle|DPendulum|Hopfield|LCoupled|ecgca115|ecgca515|
> |-|-|-|-|-|-|-|-|
> | Additive |0.61|0.68 | 0.95 |0.64|0.71|0.69|0.21|
> | Concatenate|0.49|0.53|0.92|0.4|0.57|0.68|0.20|

---

> > ### Author Rebuttal · Reviewer_fAnw · 2026-04-03
> >
> > I thank the authors for their time. However, I must say that their response does not address my concerns.
> >
> > W2/Q1: My concerns was the theoretical justification. Empirical results do not explain why the frequency structure aligns across spaces. The theoretical gap remains.
> >
> > W3/Q2: A module that actively harms performance when used independently raises fundamental questions about its design soundness. Requiring another module to compensate for introduced degradation is not a compelling justification.
> >
> > W4/Q3: The authors concede the absence of explicit directed information flow. However, shared down-projection constitutes parameter sharing, not channel dependency modeling. The gap between the claimed functionality and actual mechanism remains. Like the Frequency Adapter, this module is simply a combination of existing components.
> >
> > W5/Q4: Its difficult to accept. Multichannel complexity is a central contribution of this paper, yet most experiments are limited to datasets with only 3–6 channels. It is unclear why a method specifically designed to address multichannel complexity would exclude more channels datasets from its scope. Furthermore, high-dimensional typically refers to the number of features exceeds the number of samples. For deep learning datasets with thousands of samples, 21 channels or even a few hundred channels can hardly be considered high-dimensional. This significantly affacts the quality of the paper.
> >
> > As the authors are trying to develop some theoretical arguments, I must ask for rigor. The Kolmogorov-Szegő formula assumes stationarity and Gaussianity, yet the experimental datasets are non-stationary and non-Gaussian. The local approximate stationarity justification is insufficient. Similarly, Fano's inequality bounds error probability for discrete classification. It remains unjustified to extend it to continuous forecasting requirements.
> >
> > I maintain my recommendation of weak reject. Both adapter modules are simply combinations of existing components, the channel adapter does not implement the cross-channel modeling it claims. The theoretical assumptions is disconnect to actual data characteristics. These gaps are not resolved by the rebuttal.

---

> > > ### Author Response · Authors · 2026-04-06
> > >
> > > Thank you for giving us the opportunity to provide a detailed response.
> > >
> > > >{W2/Q1} Theoretical gap for topK on embeddings and raw data.
> > >
> > > As a heuristic theoretical motivation, we make a weaker claim: **dominant temporal-band ranking can be approximately preserved in latent space up to frequency-dependent reweighting and limited leakage**. Let $z=f(x)$ denote the TSFM embedding sequence, and consider a perturbation $\Delta x$ around a fixed input window. We locally linearize the frozen encoder as $\Delta z:= f(x+\Delta x)−f(x) \approx J_x \Delta x$. This local approximation characterizes only how small input perturbations propagate through the encoder, rather than the full spectrum of the embedding sequence $z$ itself.
> > >
> > > If the induced temporal mixing of the local Jacobian $J_x$ is approximately shift-structured along time, then the Fourier basis approximately diagonalizes the operator up to limited cross-band leakage, yielding $\Delta \hat{z}(\omega) \approx H_x(\omega) \Delta \hat{x}(\omega)$. Note that $\Delta \hat{x}(\omega)$ and $\Delta \hat{z}(\omega)$ are the temporal Fourier coefficients of $\Delta x$ and $\Delta z$ and $H_x(\omega)$ denotes the corresponding effective local frequency response induced by $J_x$.
> > >
> > > Hence the latent perturbation spectral energy, $S_{\Delta z}(\omega):=|\Delta \hat{z}(\omega)|^{2}$, can be viewed as an approximately reweighted version of the raw perturbation spectral energy $S_{\Delta x}(\omega):=|\Delta \hat{x}(\omega)|^2$: $S_{\Delta z}(\omega) \approx |H_x(\omega)|^2 S_{\Delta x}(\omega)$.
> > >
> > > Under sufficiently limited cross-band leakage and sufficiently moderate variation of $|H_x(\omega)|^2$ across nearby bands, a clear spectral gap in the raw perturbation spectrum implies that the ranking of dominant temporal bands can remain approximately stable after encoding. This should be read only as a local heuristic for dominant-band stability under the encoder response. Since Eq. (5) applies FFT to backbone embeddings rather than to the raw sequence, our claim concerns **coarse dominant-band ranking**, not one-to-one preservation of exact raw frequency bins.
> > >
> > > >{W3/Q2} Frequency adapter-only's degradation.
> > >
> > > In this paper, we do not propose selecting and using only one adapter when a certain complexity is low. This is because both the temporal complexity measure and its theoretical motivation **assume computations carried out for each individual channel**. Therefore, Time-PEFT is intentionally designed to jointly use the frequency and channel adapters, with the channel adapter providing channel-specific up-projections. To make this point clearer, we will revise the main paper to emphasize more explicitly that the full Time-PEFT should be considered and used together.
> > >
> > > >{W4/Q3} Cross-channel operations.
> > >
> > > The channel adapter uses a shared projection together with channel-specific projections, so the shared projection is designed to **extract common information across channels** that enable cross-channel operations. The empirical conclusion remains that **channel adapter is consistently useful on the complex datasets in Table 4.**
> > >
> > > >{W5/Q4} high-dimensional datasets.
> > >
> > > We sincerely thank the reviewer for the importance of datasets with several **hundred** dimensions. We conducted experiments on the ECL dataset (# channels = 321; temporal complexity = 0.294; multichannel complexity = 0.259), as it is a high-dimensional and complex dataset. As a result, **we found that Time-PEFT achieves better MSE performance than LoRA on high-dimensional datasets**.
> > >
> > > Table. Average MSE on MOMENT$_{base}$ with horizon={96, 192, 336}.
> > > ||ECL|
> > > |-|-|
> > > |LoRA| 0.2015|
> > > |Time-PEFT|**0.1967**|
> > >
> > > >{+@} Kolmogorov-Szegő formula's assumptions.
> > >
> > > We would like to respectfully clarify that, as indicated by **our section titles "theoretical motivation" in Sections 4.3.1 and 4.4.1**, the purpose of our analysis was not to provide a formal justification, but rather to offer theoretical motivation together with empirical alignment evidence. In other words, our goal was to provide **design intuition and a motivating principle**. Accordingly, in Appendix B.1.2, we discussed the possibility of presenting the idealized assumptions of stationarity and Gaussianity based on Dahlhaus (1997) and Cover (1999).
> > >
> > > >{+@} Discreteness of Fano's inequality.
> > >
> > > We use $H$ for discrete entropy and $h$ for differential entropy rate throughout both the main paper and the appendix to distinguish the two notions clearly. Accordingly, for the **multichannel complexity**, we compute the $H(\cdot)$ terms required for transfer entropy using **quantile-based discretization**, which is a commonly used practical choice. Based on this formulation, we introduced Fano’s inequality as the corresponding theoretical motivation for the multichannel complexity. However, we agree that this point was not explained clearly enough, and we will revise both the main paper and the appendix to make this distinction and motivation more explicit.

---

### Official Review · Reviewer_gvLC · 2026-03-04

**Soundness:** 3
**Presentation:** 3
**Significance:** 3
**Originality:** 3
**Overall Recommendation:** 4
**Confidence:** 4

**Summary:**

This paper proposes Time-PEFT, a parameter-efficient fine-tuning framework for time-series foundation models on complex datasets. The authors introduce two entropy-based complexity metrics—temporal complexity (spectral entropy) and multichannel complexity (transfer entropy)—to identify challenging datasets and selectively augment LoRA with a frequency adapter (using FFT-based top-k filtering) and a channel adapter (using shared and channel-specific projections).

Experiments across five TSFMs and seven complex datasets show Time-PEFT yields consistent gains over strong fine-tuning baselines, achieving up to 2.51× better MSE than LoRA on complex datasets under specific settings, while adding limited trainable parameters. Ablation studies confirm the two adapters work synergistically.

**Compliance With Llm Reviewing Policy:**

Affirmed.

**Final Justification:**

This work is novel and solid. The rebuttal addressed all my concerns. I would support the acceptance of this paper.

**Key Questions For Authors:**

1. The channel adapter dimensions in Equation 6 (Section 4.4.2) may need clarification. With W^(S)^T[E^(c)_back || E^(c)_filt] producing an r-dimensional vector, the multiplication by W^(c) ∈ R^{r×h1} seems unclear to me. I may be misunderstanding the notation, but would W^(c) ∈ R^{h1×r} align better with standard matrix multiplication?
2. The thresholds (temporal complexity >0.6, multichannel complexity >0.1) for defining "complex" datasets appear heuristic. How were they chosen, and how portable are they across datasets/backbones?
3. While I understand Figures 3 and 8 aim to demonstrate that complex datasets exhibit high variance, the visualization is unclear to me. The statement "complex datasets exhibit high variance" and "zero-shot results appear as outliers" requires clarification. In Figure 8(a), ECGCA515 shows multiple outliers—do all represent zero-shot performance, or do some fine-tuning methods also produce outliers? Please clarify the boxplot construction and explicitly label which points correspond to zero-shot vs. fine-tuning methods.

**Limitations:**

yes

**Strengths And Weaknesses:**

Strengths

Uses information-theoretic metrics (spectral entropy and transfer entropy) to operationalize “complexity” in forecasting and to motivate which aspects to adapt (temporal vs. inter-channel).

Experiments are comprehensive, evaluating five TSFMs across seven complex datasets (chaotic and medical) plus standard benchmarks (appendix), with meaningful ablations isolating adapter contributions and parameter analysis showing modest overhead. Code is released for reproducibility.

The high-level motivation, complexity axes, and architecture are clearly communicated. Each adapter's role is well-articulated, with design choices effectively connected to the proposed temporal and multichannel complexity measures.

Weaknesses

Although MSFT and AdaPTS are cited in the introduction, compared in Table 1 regarding channel/temporal capabilities, they are notably absent from experimental baselines (Section 5.1.3, Tables 2-10) without explanation.

The “top-k” frequency filtering is global and hard-coded; no mechanism adapts k to dataset complexity or to per-channel variability, and frequency leakage/aliasing issues are not addressed.

The paper appears to report single-run results without mentioning multiple trials, random seeds, or error bars, which limits confidence in the statistical significance of the reported improvements. While understandable given the substantial experimental workload, this could be addressed in the camera-ready version if accepted after rebuttal.

---

> ### Author Rebuttal · Authors · 2026-03-31
>
> We are grateful for the reviewer’s constructive suggestions.
>
> >W1. No MSFT and AdaPTS.
>
> Thank you for asking additional baselines. We conducted experiments on **AdaPTS and MSFT in Table D**. Time-PEFT achieves better performance than the compared methods.
>
> Table D. Average MSE on MOMENT$_{base}$ and UniTS with horizon={96,192,336}.
> |TSFM|Baseline |Lorenz|CellCycle|DPendulum|Hopfield|LCoupled|ecgca115|ecgca515|
> |-|-|-|-|-|-|-|-|-|
> |MOMENT|MSFT|1.04|0.94|1.01|0.92|1.14|1.14|0.36|
> ||AdaPTS|1.10|0.99|1.08 |0.91|1.18|1.11|0.40|
> ||**OURS**|**0.49**|**0.53**|**0.92**|**0.48**|**0.57**|**0.68**|**0.20**|
> |UniTS|MSFT|0.48 |0.44|0.94|0.41|0.62|0.57|0.17|
> ||AdaPTS|1.20|1.07|1.09|1.20|1.24|1.16|0.40|
> || **OURS** |**0.16**|**0.14**|**0.85**|**0.19**|**0.26**|**0.54**|**0.13**|
>
> > W2. No adaptive top-k mechanism.
>
> Because FFT is applied on a **per-channel basis within each batch**, we did not introduce an additional adaptive mechanism for top-k selection to account for per-channel variability. Moreover, as shown in Figure 7, the performance is not highly sensitive to changes in top-k, and thus we did not adopt an adaptive top-k mechanism in the current version. Approaches such as applying windowing before FFT to mitigate leakage, or selecting adaptive top-k values for each dataset, are very promising directions for future work. We greatly appreciate this helpful suggestion.
>
> > W3. No standard deviation(std).
>
> In the current experiments, we used the default random seed 0. We **additionally ran experiments** with seeds 1 and 2 and were able to obtain partial results **in Table E**. Even when extended to multiple seeds, Time-PEFT still consistently outperforms LoRA. We will include results that take the std into account in the final version.
>
> Table E. Average MSE on MOMENT$_{small}$ with horizon={96,192,336}.
>
> ||| Lorenz|CellCycle|DPendulum|Hopfield|LCoupled|ecgca115|ecgca515|
> |-|-|-|-|-|-|-|-|-|
> |LoRA|Avg | 0.67 | 0.76 | 0.99 | 0.72 | 0.88 | 0.80 | 0.27 |
> || Std | 0.13 | 0.13 | 0.05 | 0.12 | 0.13 | 0.19 | 0.05 |
> |**Time-PEFT** | Avg | **0.56** | **0.60** | **0.94** | **0.54** | **0.64** | **0.69** | **0.20** |
> || Std | 0.15 | 0.18 | 0.06 | 0.14 | 0.17 | 0.18 | 0.05 |
>
> > Q1. Would $W^{(c)} \in \mathcal{R}^{[h1×r]}$ align better?
>
> Thank you very much for carefully checking the equation. As you pointed out, "$W^{(c)} \in \mathcal{R}^{[h1×r]}$" is the correct form. We also rechecked the implementation and confirmed that it follows this correct formulation. We will fix the error in the final version.
>
>
> > Q2. Complexity thresholds.
>
> We agree that the complexity thresholds are **empirical** values. Note that these thresholds are derived by analyzing the datasets themselves **without involving backbone models**. As shown in Figure 2, standard datasets cluster below 0.6 in temporal complexity and below 0.1 in multichannel complexity, while complex datasets consistently lie above at least one of these thresholds. As can also be seen in Figures 3 and 8, these thresholds effectively identify datasets on which zero-shot forecasting performs poorly, **regardless of the specific TSFM**, when used to distinguish complex from standard datasets.
>
>
> > Q3. Please clarify the boxplot construction.
>
> In Figures 3 and 8, each boxplot contains the average MSE for each dataset. As stated in the caption of Figure 8, these average MSE values are derived from "zero-shot results and existing four fine-tuning methods (forecast head fine-tuning, full fine-tuning, LoRA, and FourierFT)." As suggested, in the final version we will **explicitly state that the zero-shot results correspond to the worst MSE for each dataset.** We will also revise the sentence to: "For complex datasets, the zero-shot MSE is significantly higher in most cases, appearing as **upper** outliers relative to fine-tuning results." This will make it clearer that the upper outliers are our main point of interest. In addition, we will further clarify the notion of high variance with a concrete example. For instance, we will add a sentence: "The standard deviation for Lorenz (a complex dataset) under MOMENT is 0.2695, whereas that for ETTh2 (a standard dataset) is 0.0285."

---

> > ### Author Rebuttal · Reviewer_gvLC · 2026-04-01
> >
> > I thank the authors for their detailed response, which addresses my concerns. I will maintain my score and continue to support the acceptance of this work.

---

> > > ### Author Response · Authors · 2026-04-04
> > >
> > > We sincerely appreciate your thoughtful follow-up and are very pleased to hear that our responses fully addressed your concerns. Thank you for taking the time to carefully review both our manuscript and rebuttal. We are also truly grateful for your continued support for the acceptance of our work.

---

### Official Review · Reviewer_UhRB · 2026-03-06

**Soundness:** 3
**Presentation:** 2
**Significance:** 2
**Originality:** 3
**Overall Recommendation:** 5
**Confidence:** 4

**Summary:**

This paper proposes Time-PEFT, a parameter-efficient fine-tuning (PEFT) framework for time-series foundation models (TSFMs). Two data-driven complexity metrics are introduced: temporal complexity, derived from spectral entropy to quantify how uniformly signal energy is distributed across the frequency domain, and multichannel complexity, computed via transfer entropy to measure directed information flow between channels. These metrics are used to identify "complex" datasets where zero-shot TSFM performance is poor. Guided by this diagnosis, Time-PEFT integrates a frequency adapter (top-k FFT filtering applied to backbone embeddings) and a channel adapter (shared down-projection plus channel-specific up-projections) into existing PEFT backbones such as LoRA. Theoretical motivation draws on the Kolmogorov-Szego formula and Fano's inequality to justify each adapter. Experiments across five TSFM backbones and seven complex datasets demonstrate consistent improvements over standard fine-tuning baselines, with the strongest gains on MOMENT and TTM.

**Compliance With Llm Reviewing Policy:**

Affirmed.

**Final Justification:**

The paper makes a meaningful contribution and the rebuttal has adequately addressed my concerns. The overall quality is solid and the work is ready for acceptance.

**Key Questions For Authors:**

Q1: Does the shared down-projection in the channel adapter include an activation function sigma(.)? If yes, please add it to Equation 6 and specify its type (e.g., ReLU, layer norm). If no, please correct Algorithm 1 line 14. The inconsistency makes it impossible to determine the actual implemented model structure, which is fundamental to reproducing the results.

Q2 : Please explicitly state in the main text that Time-PEFT uses only the Chronos encoder while the five standard baselines use the full encoder-decoder. Additionally, please provide Time-PEFT results with the complete encoder-decoder architecture on complex datasets (currently Table 10 only covers standard datasets). Without this, the fairness of Chronos results in Table 2 cannot be assessed.

Q3: What specific value of L is used in Definition 3.2 (T = {1, ..., L})? Is it set to the look-back window (96) or the forecast horizon? Please provide complete computation details for the complexity metrics to enable reproducibility.

**Limitations:**

YES

**Strengths And Weaknesses:**

Strengths:

1. The theoretical framework is internally consistent. The Kolmogorov-Szego formula provides a principled link between spectral peakiness and prediction error reduction, and the Fano inequality derivation cleanly formalizes the benefit of multichannel modeling in terms of transfer entropy.
2. The motivating experiment design is well thought out. Defining complexity along two orthogonal axes (temporal and channel) and separately addressing each with a dedicated adapter is a coherent and principled approach.
3. The comparison against extended single-seed LoRA training is absent, but the broad backbone coverage (five different TSFMs) and the dataset diversity (five chaotic systems plus two medical ECG datasets) provide meaningful generalization evidence.

Weaknesses:

1. Algorithm 1 (Appendix C, line 14) introduces an activation function sigma(W^(S)*Z^(c)) in the shared down-projection step, while Equation 6 (Section 4.4.2) writes E_ch^(c) = W^(c) * W^(S)^T * [E_back^(c) || E_filt^(c)] without any activation. These two descriptions of the same operation are inconsistent, making it impossible to determine the actual implemented model structure.
2. The Chronos comparison in Table 2 is architecturally asymmetric. As disclosed in Appendix D.1.2, Time-PEFT and ChannelMixing use only the encoder of Chronos, while HeadOnly, FullFT, LoRA, and FourierFT preserve the full encoder-decoder structure. This architectural mismatch is not mentioned in Section 5.1.2 or Section 5.2. Furthermore, encoder-decoder results for Time-PEFT on complex datasets are never reported (Table 10 only covers standard datasets), leaving the fairness of the main Chronos results unresolved.
3. The complexity thresholds of 0.6 (temporal) and 0.1 (multichannel) are defined without justification. No theoretical derivation, ablation over threshold values, or sensitivity analysis is provided.
4. The look-back window length L in Definition 3.2 (T = {1, ..., L}) is never specified anywhere in the paper. For chaotic systems, the choice of lag tau critically affects transfer entropy values, and this omission undermines the reproducibility of the complexity metrics.

---

> ### Author Rebuttal · Authors · 2026-03-31
>
> We sincerely thank for the careful and detailed review.
>
> > {W1+Q1}. No $\sigma$ in Eq 6.
>
> Thank you for carefully examining the equation. In the Time-PEFT implementation provided via the anonymous link in the main paper, activation functions (ReLU and dropout) are applied. In the final version, we will revise Eq. 6 to **explicitly include the activation function $\sigma$**, and we will also **describe in detail which activation functions are used**.
>
>
> > {W2+Q2}. Time-PEFT and ChannelMixing in Chronos was evaluated using an encoder-based approach rather than an encoder-decoder-based approach.
>
> Time-PEFT is designed to add a frequency adapter and a channel adapter on top of high-quality embeddings extracted from pre-trained TSFMs, followed by a forecast head. Accordingly, we conducted experiments in an encoder-based setting to fine-tune Time-PEFT using embeddings extracted from the pre-trained encoder. For this reason, we stated that "encoders are utilized as backbones". In particular, for the encoder-decoder model Chronos, using the encoder as the backbone is necessary to properly obtain the effect of the proposed method. In Chronos, the encoder outputs embeddings of shape (batch_size\*channel_size, patched_seq_len, d_model), whereas the decoder outputs embeddings of shape (batch_size\*channel_size, 1, d_model). Therefore, the output from the decoder cannot be used with the frequency adapter.
>
> In the main paper, when applying existing PEFT baselines, we attempted to preserve the original TSFM structures and pre-trained weights as much as possible. However, as pointed out, **we will revise the experiments to ensure that all baselines and backbones in Chronos are evaluated under the same encoder-only setting**. Accordingly, as shown in Table C, when baselines are evaluated using encoder-only Chronos, Time-PEFT achieves better performance than LoRA across all datasets and also outperforms other baselines. **We will update the Chronos baselines in the main paper accordingly and revise the corresponding descriptions**.
>
> Table C. Average MSE on encoder-based Chronos with horizon={96,192,336}.
> ||Lorenz|CellCycle|DPendulum|Hopfield|LCoupled|ecgca115|ecgca515|
> |-|-|-|-|-|-|-|-|
> |HeadOnly| 0.74 | 0.80 | 0.97 | 0.70 | 0.86 | 0.89 | 0.32 |
> |Full FT| 0.37 | 0.51 | **0.86** | 0.45 | 0.53 | 1.10 | 0.51 |
> |LoRA| 0.57 | 0.78 | 0.96 | 0.64 | 0.74 | 1.19 | 0.64 |
> |FourierFT| 0.74 | 0.80 | 0.97 | 0.70 | 0.86 | 0.89 | 0.32 |
> |ChannelMixing| 0.56 | 0.60 | 0.95 | 0.56 | 0.64 | 0.75 | 0.27 |
> |Time-PEFT| **0.28** | **0.44** | 0.87 | **0.39** | **0.44** | **0.72** | **0.24** |
>
>
> > W3. No theoretical derivation of threshold.
>
> We agree that the complexity thresholds are empirically determined. As shown in Figure 2, standard datasets cluster below 0.6 in temporal complexity and below 0.1 in multichannel complexity, while complex datasets consistently lie above at least one of these thresholds. Since these are not theoretically derived thresholds, **we designed the complexity metrics in a conservative manner.**
>
> For example, when computing quantities such as transfer entropy, the results can be highly sensitive to the choice of time lag. Therefore, instead of using a single time lag, we construct the multichannel complexity using the maximum value over multiple time lags. In this process, **rather than selecting an arbitrary single lag, we use a commonly adopted look-back window (=96) so that the complexity metric does not depend on a specific lag choice.**
>
>
> > {W4+Q3}. Look-back window length L in Def 3.2.
>
> Thank you for the careful reading. In our experiments, the look-back window is set to 96, consistent with the forecasting setup. To compute multichannel complexity, the time lag $\tau$ is evaluated for all values from 1 to **L=96**, and **all datasets are evaluated under the same setting**. To improve clarity, we will revise the description as follows: "we define the final score as the average of the maximum TE values over all time lags in $\mathcal{T} = \{1, \dots, L\}$, **where L is the look-back window length.**" We will also explicitly add the following statement: "In our experiments, the look-back horizon was fixed to $L = 96$."

---

> > ### Author Rebuttal · Reviewer_UhRB · 2026-04-02
> >
> > The detailed rebuttal is appreciated. I am willing to raise my score.

---

> > > ### Author Response · Authors · 2026-04-04
> > >
> > > We sincerely thank you for the thoughtful follow-up and for your positive assessment of our rebuttal. We are glad that our responses have adequately addressed your concerns, and we truly appreciate your willingness to raise your score.

---

### Official Review · Reviewer_gq69 · 2026-03-12

**Soundness:** 4
**Presentation:** 4
**Significance:** 4
**Originality:** 4
**Overall Recommendation:** 5
**Confidence:** 4

**Summary:**

Time-PEFT introduces a novel finetuning framework specifically for time-series foundation models (that do time series forecasting), arguing that standard finetuning such as LoRA is insufficient for complex time series datasets. The paper introduces multiple theoretical measures of complexity, which motivate a new finetuning method using two adapters, a channel adapter and a frequency adapter. The results show improvements across five TSFMs and seven datasets of varying complexity, demonstrating improvements over various finetuning methods, including head only, full, LoRA, FourierFT, and ChannelMixing.

**Compliance With Llm Reviewing Policy:**

Affirmed.

**Key Questions For Authors:**

1. Does Time-PEFT maintain strong performance in long-horizon forecasting, e.g. for a horizon of length 720?
2. In terms of performance, do the channel/frequency adapters similarly improve MSE on top of head-only finetuning, or even full finetuning? This would help isolate the effect of LoRA on the method.
3. Do the predictions offered by the finetuning offer better interpretation of the predictions than other finetuning methods? It would be useful to see case studies.
4. How sensitive are rank and top-k in lower regimes, and how should one pick them when applying Time-PEFT?

**Limitations:**

Yes

**Strengths And Weaknesses:**

Strengths
- The finetuning problem for TSFMs is very important for the community, and Time-PEFT demonstrates this over complex datasets and various TSFMs.
- The paper is well-written and easy to read.
- Time-PEFT improves over previous attempts at time-series finetuning methods that only cover either channel or temporal domain improvements.
- The mapping between channel/temporal complexity and finetuning performance is very interesting and the theoretical explanation is insightful.

Weaknesses
- Time-PEFT underperforms on general or standard datasets, which prevent it from being fully general. This forces the user to choose when to use one method or the other.
- Time-PEFT builds on top of/is not independent of LoRA, making its parameter efficiency strictly worse than LoRA.
- It is unclear how to choose hyperparameters for Time-PEFT across datasets, as MSE worsens as rank/top-k goes up, and results are not given for small rank/top-k.

---

> ### Author Rebuttal · Authors · 2026-03-31
>
> We appreciate the positive perspective and helpful feedback on our work.
>
> > W1. Time-PEFT underperforms on standard datasets. Time-PEFT forces the user to choose when to use one method or the other.
>
> Time-PEFT is designed to **improve performance on complex datasets where there is a large gap between zero-shot and fine-tuning performance.** To this end, we propose empirical metrics—temporal complexity and multichannel complexity—along with corresponding thresholds to define what constitutes a complex dataset. Accordingly, users can apply Time-PEFT (OURS) when the dataset satisfies temporal complexity ≥ 0.6 or multichannel complexity ≥ 0.1. In other words, **by using the provided thresholds, users can determine which fine-tuning method to use without significant difficulty.**
>
>
> > W2. Time-PEFT's parameter efficiency is strictly worse than LoRA.
>
> When fine-tuning complex datasets using MOMENT, Time-PEFT achieves up to approximately a maximum 40% performance improvement over LoRA. Therefore, we believe that the **additional overhead in the number of trainable parameters is acceptable given the substantial performance gains**.
>
>
> > {W3.+Q4.} Hyper-parameter sensitivity in lower regimes.
>
> We appreciate the opportunity to provide additional experiments on lower regimes. We additionally provide hyperparameter sensitivity results for **lower regimes (lower ranks: 2, 4, 6 and lower top-k: 3, 6, 9)** in [Figure A🔗](https://anonymous.4open.science/r/TimePEFT/asset/FigureA.pdf) on MOMENT$_{base}$ with horizon={96,192,336}. The results show that the performance differences are not significant in the low regime. For the top-k parameter, this is because the channel adapter receives not only the filtered embedding but also the original embedding. For the rank parameter, it does not contribute significantly to the number of trainable parameters in lower regimes. Therefore, based on our current setting, both hyperparameters are relatively stable.
>
>
>
> > Q1. long-horizon forecasting (720).
>
> To evaluate long-horizon performance, we provide results **with horizon = 720 in Table A**. The results show that Time-PEFT achieves the best performance on most datasets.
>
> Table A. MSE on MOMENT$_{base}$ with horizon=720.
> || Lorenz | CellCycle | DPendulum | Hopfield | LCoupled | ecgca115 | ecgca515 |
> |-|-|-|-|-|-|-|-|
> |Zeroshot|1.25|1.19|1.13|1.22|1.37|1.51|0.47|
> |HeadOnly|0.87|0.98|1.05|0.93|1.08|1.24|0.38|
> |FullFT|1.23 |1.15 |**1.03** |0.93 |**0.94**|1.49|0.43|
> |LoRA|0.87|0.98|1.05|0.93|1.08|1.24|0.38|
> |FourierFT|0.87|0.98|1.05|0.93|1.08|1.24|0.38 |
> |ChannelMixing|1.00|1.02|1.10|0.97|1.10|1.17|0.36|
> |**Time-PEFT**|**0.84**|**0.94**|**1.03**|**0.87**|0.95|**1.16**|**0.35**|
>
>
> > Q2. Ablation study on the forecast head.
>
> The proposed method trains the frequency adapter, channel adapter, and forecast head based on embeddings from a backbone augmented with LoRA. However, as suggested, we also conducted experiments **by removing the LoRA component from the backbone.** In Table B, the results show that the **proposed architecture (F_both+C) still achieves the best** performance on most datasets.
>
> Table B. Average MSE on MOMENT$_{base}$ with horizon={96, 192, 336}. Note that H is a headonly **(no LoRA)**, C is a channel adapter, and F is a frequency adapter. We distinguish two variants of F: F_only transmits only the filtered embedding to the subsequent module, whereas F_both propagates both the filtered and backbone embeddings.
> || Lorenz | CellCycle | DPendulum | Hopfield | LCoupled | ecgca115 | ecgca515 |
> |-|-|-|-|-|-|-|-|
> |H| 0.60 | 0.69 | 0.97 | 0.66 | 0.80 | 0.79 | 0.26 |
> |H+C| 0.58 | 0.65 | 0.94 | 0.59 | 0.67 | **0.68** | 0.20 |
> |H+F_only| 1.19 | 1.05 | 1.08 | 1.12 | 1.22 | 1.14 | 0.40 |
> |H+F_both| 0.69 | 0.78 | 0.99 | 0.74 | 0.90 | 0.80 | 0.26 |
> |**H+F_both+C**| **0.57** | **0.62** | **0.94** | **0.55** | **0.65** | 0.69 | **0.20** |
>
>
> > Q3. Case Studies.
>
> Thank you for the suggestion regarding case studies. As suggested, we provide visualizations for MOMENT$_{small}$ in [Figure B🔗](https://anonymous.4open.science/r/TimePEFT/asset/FigureB.pdf). From these case studies, we observe that Time-PEFT effectively captures diverse patterns across datasets. In particular, compared to LoRA, it **better captures periodic patterns and sharp increasing trends**. In the final version, we will include additional case study results across more TSFMs.

---

> > ### Author Rebuttal · Reviewer_gq69 · 2026-04-01
> >
> > Thanks to the authors for the detailed followup. I maintain my score.

---

> > > ### Author Response · Authors · 2026-04-04
> > >
> > > We are grateful for your follow-up and for confirming that our responses have fully addressed your concerns. Thank you for your careful evaluation and for your time throughout the review process.

---

### Decision · Program_Chairs · 2026-04-30

**Decision:**

Accept (regular)

**Comment:**

Reviewers agreed that this paper is written clearly and well-motivated. The theoretical framework is interesting, and the idea of introducing data-driven metrics to quantify temporal and multichannel complexity is insightful. The overall experimental performance is also good. During the rebuttal, most of the reviewers’ concerns are resolved. Some remaining concerns include: (1) the proposed modules are based on existing components; (2) the relationship between parameter shared projection and explicit cross-channel operations. We encourage authors to pay attention to these points during revision.